# Assessing London $CO_2$, $CH_4$ and CO emissions using aircraft measurements and dispersion modelling

Joseph R. Pitt[1], Grant Allen[1], Stéphane J.-B. Bauguitte[2], Martin W. Gallagher[1], James D. Lee[3], Will Drysdale[3], Bethany Nelson[3], Alistair J. Manning[4], Paul I. Palmer[5]

[1]School of Earth and Environmental Sciences, University of Manchester, Oxford Road, Manchester, M13 9PL, UK
[2]Facility for Airborne Atmospheric Measurements (FAAM), Cranfield University, Cranfield, MK43 0AL, UK
[3]Wolfson Atmospheric Chemistry Laboratories, Department of Chemistry, University of York, Heslington, York YO10 5DD, UK
[4]Met Office, FitzRoy Road, Exeter, EX1 3PB, UK
[5]School of GeoSciences, University of Edinburgh, Edinburgh, EH9 3FF, UK

*Correspondence to:* Joseph Pitt (joseph.pitt@manchester.ac.uk)

**Abstract.** We present a new modelling approach for assessing atmospheric emissions from a city, using an aircraft measurement sampling strategy similar to that employed by previous mass balance studies. Unlike conventional mass balance methods, our approach does not assume that city-scale emissions are confined to a well-defined urban area and that peri-urban emissions are negligible. We apply our new approach to a case study conducted in March 2016, investigating CO,

$CH_4$ and $CO_2$ emissions from London using aircraft sampling of the downwind plume. For each species, we simulate the flux per unit area that would be observed at the aircraft sampling locations based on emissions from the UK national inventory, transported using a Lagrangian dispersion model. To reconcile this simulation with the measured flux per unit area, assuming the transport model is not biased, we require that inventory values of CO, $CH_4$, and $CO_2$ are scaled by 1.00, 0.70, and 1.57, respectively. However, our result for $CO_2$ should not be considered a direct comparison with the inventory which

only includes anthropogenic fluxes.

For comparison, we also calculate fluxes using a conventional mass balance approach and compare these to the emissions inventory aggregated over the Greater London area. Using this method we derive much higher inventory scale factors for all three gases, as a direct consequence of the failure to account for emissions outside the Greater London boundary. That substantially different conclusions are drawn using the conventional mass balance method demonstrates the

danger of using this technique for cities whose emissions cannot be separated from significant surrounding sources.

## 1. Introduction

Over half the people in the world (54%) live in urban areas. This proportion is projected to increase to 66% by 2050 (United Nations, 2014). Consequently, cities are responsible for a large proportion of anthropogenic greenhouse gas (GHG) emissions. The 2015 UNFCCC Paris Agreement requires signatory states to not only report national GHG emissions, but

also to establish and improve independent methods for verifying these reported emissions (UNFCCC, 2015). Top-down

methods that use atmospheric measurements to determine city-scale emissions can assess the accuracy of bottom-up emission inventories and provide crucial information to help improve bottom-up accounting methods.

In the UK, spatially and sectorally disaggregated emissions calculated using a bottom-up methodology are given in the National Atmospheric Emissions Inventory (NAEI; Brown et al., 2017). For Greater London, nearly all sources of anthropogenic $CO_2$ and CO emissions are associated with fuel combustion. For $CO_2$, the main sources are domestic and commercial combustion and road transport, while emissions from power stations are largely located outside the Greater London administrative boundary. Emissions of CO are comprised of a range of combustion sources, with road transport emissions constituting the largest category. For $CH_4$ the principal sources are waste treatment and disposal, and leakage during natural gas distribution (in contrast to the UK as a whole, where emissions associated with ruminant livestock dominate). Providing a top-down constraint on these London emissions is important, not only because London represents a large emission source in its own right, but also because it can help inform the assumptions that go into calculating bottom-up emission estimates for these sectors at a national level.

Natural emissions, which are not included in the NAEI, contribute to varying extents for the three species. For $CH_4$ wetlands are the most significant source of natural emissions within the UK, but wetland fluxes from London and its surrounding areas are negligible compared to anthropogenic emissions. Likewise, in urban areas CO is dominated by anthropogenic sources, although oxidation of biogenic VOCs can contribute, especially during the summer. However, for $CO_2$ biospheric fluxes not included in the inventory have a significant impact on measured mole fractions downwind of the UK; the impact of these fluxes is discussed further in Sect. 3.1.2.

Aircraft mass balance techniques have previously been employed to measure trace gas fluxes from several cities (e.g. Mays et al., 2009; Turnbull et al., 2011; Cambaliza et al., 2014) including London (O'Shea et al., 2014a). Typically, horizontal transects are conducted downwind of a city at several altitudes to sample its emitted plume for various trace gas species. These vertically-stacked transects define a 2D plane of sampling downwind of the city. A background mole fraction can be determined by sampling upwind of the city or from downwind measurements outside of the plume. The mass flux of the plume through the 2D plane of sampling is then calculated from the measured mole fraction enhancements (above background) and wind speed.

Using the mass balance method, it is relatively simple to determine the flux for a given species through this 2D sampling plane, relative to a defined background mole fraction. However, attributing this calculated value to a given city, region or emission source can be far more difficult. In the case of an isolated city, whose surrounding areas can be considered comparatively negligible emission sources, the flux calculated using the mass balance method can be considered representative of the sum total emissions from the city. This enables a direct comparison to be made between top-down and bottom-up emission estimates for the city as a whole. However, for cities such as London, which is surrounded by other non-negligible emission sources for $CO_2$, $CH_4$ and CO, it is difficult to associate the flux derived using a conventional mass balance approach with a well-defined spatial region. This in turn makes it hard to directly compare top-down and bottom-up

emission estimates, as it is not clear how to unambiguously determine the area over which bottom-up emissions should be aggregated to enable comparison with top-down fluxes.

Another approach involves the use of an atmospheric transport model to represent transport of emitted species from source to measurement site. This enables simulated enhancements to be calculated at the measurement location based on a prescribed emissions map (e.g. from a bottom-up inventory). A range of inverse modelling techniques can then be employed to optimise the emissions map according to the measured enhancements. This is frequently performed at a regional scale using ground-based measurements from long-term monitoring sites (e.g. Manning et al., 2011; Bergamaschi et al., 2015; Ganesan et al., 2015), but it has also been performed using aircraft data to provide the spatial sampling coverage required to estimate city-scale emissions using a few hours of measurement data (e.g. Brioude et al., 2013). One of the key challenges associated with this approach is to control the behaviour of the inversion without over constraining it such that the results only reflect the choice of prior emissions. This is particularly difficult using aircraft data, as for an individual flight the information content of the dataset is limited in comparison to long-term ground-based measurements.

In this study we have developed an alternative method for assessing bottom-up inventory fluxes, using the same aircraft sampling technique typically employed by mass balance studies. In this new approach we have used the UK Met Office's Lagrangian dispersion model, NAME (Numerical Atmospheric dispersion Modelling Environment), to simulate the transport of the inventory fluxes to the location of the measurements. Rather than optimising an emissions map by using a cost function to reduce model-measurement mismatch (as typically used in previous atmospheric transport inversion studies), we take the simple approach of comparing the average measured and simulated fluxes at the aircraft sample locations and rescale the inventory according to their ratio. This approach effectively constrains the inversion behaviour by removing the freedom to spatially redistribute emissions, while allowing it total freedom to adjust the magnitude of emissions relative to the prior. On the other hand, relative to the mass balance method, we avoid the need to aggregate inventory emissions over some arbitrarily defined area on the ground in order to compare them to calculated top-down flux values.

In addition to applying our new method, we have also applied the conventional mass balance technique to the same data, and compared the top-down fluxes derived to inventory emissions aggregated over the Greater London administrative area. The two approaches reach significantly different conclusions regarding the accuracy of the emissions inventory; we suggest that this is a consequence of sources outside Greater London that are not adequately accounted for by the background mole fraction calculation, yet nonetheless contribute to the plume measured downwind of the conurbation.

## 2. Case study details

### 2.1 Aircraft measurements and calibration

We recorded measurements on board the UK's Facility for Airborne Atmospheric Measurement (FAAM) BAe-146 atmospheric research aircraft (henceforth referred to as the FAAM aircraft). For full details of the aircraft payload see Palmer
et al. (2018). Here we describe only those measurements that are relevant to this case study.

Mole fractions of $CO_2$ and $CH_4$ were measured using a Fast Greenhouse Gas Analyser (FGGA; Los Gatos Research, USA). Paul et al. (2001) describe the operating principle of the instrument, and O'Shea et al. (2013) provide details on the instrument operational practice and performance on the FAAM aircraft across several campaigns. The FGGA was calibrated hourly in flight, using two calibration gas standards traceable to the WMO-X2007 scale (Tans et al., 2009)
and WMO-X2004A scale (an extension of the scale described by Dlugokencky et al., 2005) for $CO_2$ and $CH_4$, respectively. For this case study, the certified standards (369.54 ppm $CO_2$, 1853.8 ppb $CH_4$ and 456.97 $CO_2$, 2566.0 ppb $CH_4$ respectively) spanned the range of measured ambient mole fractions.

We measured a target cylinder containing intermediate mole fraction values approximately half way between these hourly calibrations to quantify any instrument non-linearity or drift. This flight formed part of the wider GAUGE
(Greenhouse gAs UK and Global Emissions) campaign, during which we derived average target cylinder measurement offsets of 0.036 ppm for $CO_2$ and 0.07 ppb for $CH_4$, relative to the WMO-traceable values, with standard deviations of 0.398 ppm and 2.40 ppb, respectively, for 1 Hz sampling. Each individual target cylinder measurement consisted of a 20 s sample after allowing time for the measurements to reach equilibrium. The standard deviation of these 20 s averaged values was 0.245 ppm for $CO_2$ and 1.42 ppb for $CH_4$.

Another source of measurement uncertainty was the impact of water vapour in the sampled air on the retrieved $CO_2$ and $CH_4$ mole fractions. This was principally a consequence of mole fraction dilution (i.e. an increase in the total number of molecules per unit volume relative to dry air) and pressure broadening of the spectral absorption lines. The method used to correct for this is described by O'Shea et al. (2013). Using that technique we have derived maximal uncertainties due water vapour of 0.156 ppm for $CO_2$ and 1.05 ppb for $CH_4$. Finally, there are also uncertainties associated with the certification of
the target cylinder of 0.075 ppm and 0.76 ppb, respectively. Combining all of these uncertainties with the target measurement standard deviations yields nominal total uncertainties for $CO_2$ and $CH_4$ of 0.434 ppm and 2.73 ppb at 1 Hz, and 0.300 ppm and 1.93 ppb when averaged over 20 s.

We measured CO mole fractions using vacuum ultraviolet florescence spectroscopy (AL5002, Aerolaser GmbH, Germany). The principle of this system is described by Gerbig et al. (1999), who also evaluate its performance on board an
aircraft. Calibration was performed using in-flight measurements of a single gas standard and the background signal at zero CO mole fraction. Gerbig et al. (1999) derive a 1 Hz repeatability of 1.5 ppb (at 100 ppb), and an accuracy of 1.3 ppb ± 2.4% for the 1 Hz measurements.

Details of the meteorological instrumentation on board the FAAM aircraft are provided by Petersen and Renfrew (2009). In summary, we measured temperature with a Rosemount 102AL sensor, with an overall measurement uncertainty of 0.3 K at 95% confidence; we took static pressure measurements from the air data computer, based on measurements from pitot tubes around the fuselage, with an estimated absolute accuracy of 0.5 hPa; we made 3D wind measurements using the a 5-hole probe system described by Brown et al. (1983), with an estimated uncertainty in horizontal wind measurements of < 0.5 m s $^{-1}$.

## 2.2 Sampling strategy

On 04 March 2016 we conducted a targeted case study flight to measure $CO_2$, $CH_4$ and CO mole fraction enhancements downwind of London, so as to assess the accuracy of the bottom-up emissions inventory. The flow over the region was consistently westerly, bringing background air from the northern Atlantic with an average travel time over the British Isles of 20 hours (as determined using NAME). As the influence of land-based sources on the recent history of this air mass (upwind of the British Isles) can be assumed to be negligible, we expect that air arriving at the British Isles had relatively homogeneous and well-mixed trace gas composition throughout the boundary layer prior to the influence of local fluxes. Take off was at 08:55 local time (=UTC), with the vertically-stacked transects downwind of London conducted between 11:16 and 13:32 local time. Results from NAME indicate a typical travel time between central London and the downwind sampling plane of ~5 hours, suggesting the majority of the sampled air passed over London between ~06:00 and ~09:00.

Figure 1 shows the flight track from an aerial perspective; between points A and B we flew repeated horizontal transects at various altitudes through a plume of enhanced mole fractions downwind of London emission sources. At the southernmost end of these transects, the constraints of UK airspace forced us to deviate from our desired course perpendicular to the prevailing wind. However, as we sampled the overwhelming majority of the London plume north of this imposed turning point, such that measurements during the deviation to point B represented background (out-of-plume) sampling, we do not expect this deviation to impact on the derived fluxes.

During an initial transect at 1550 m altitude we measured typical uniform free tropospheric background mole fractions for all three gases ($CO_2$, $CH_4$ and CO). Following this we descended to 120 m, and subsequently flew 6 transects of increasing altitude, breaking the final transect short to profile up to 1550 m within the observed plume. Figures 2a, 2b and 2c show these transects, coloured by mole fraction for $CO_2$, $CH_4$ and CO respectively, projected onto an altitude-latitude plane.

## 2.3 Dispersion model configuration

To determine the air history corresponding to the continuous aircraft sampling we ran the NAME dispersion model in backwards mode, releasing 100 tracer particles at each 1 Hz aircraft measurement location and tracking their motion back in time. NAME was driven by meteorological data from the UK Met Office's UKV model (Tang et al., 2013), which provides hourly data on 70 vertical levels at 1.5 km horizontal resolution over the British Isles. NAME determines particle motion based on the mean wind field (which is determined by interpolating the met data spatially and temporally to the particle

location for each time-step) and a parameterisation of unresolved turbulent and mesoscale motions (for details see Jones et al., 2007, and references therein). In this study we used a NAME model time-step of 1 minute. By way of guidance, it is worth noting that although this NAME setup is more computationally intensive than is typically employed, because the release duration was less than 3 hours and the maximum particle travel time before leaving the domain was 37 hours (and less than 24 hours for the majority of particles) the run completion time remained on the order of hours rather than days using the JASMIN scientific computing facility.

To assess the influence of surface fluxes on the sampled air, we used NAME to calculate an air history matrix for each minute of the flight (henceforth referred to as a release period), quantifying the sensitivity of the sampling to the emission in each grid square. To calculate this matrix, we recorded the total time each tracer particle spent in the lowest 100 m above ground level on a 1 km x 1 km horizontal grid (UK National Grid), matching the spatial resolution of the NAEI emissions inventory. The air history matrix $D_{ij}$ was then calculated according to:

$$D_{ij}(t) = \frac{T_{ij}(t)}{d \times n(t)} \tag{1}$$

The indices $i$ and $j$ represent the northing and easting components of the horizontal grid. $T_{ij}(t)$ is the average particle residence time in each grid box for every release period, $d$ is the height of the grid boxes (here 100 m) and $n(t)$ is the average molar density of air ($mol_{air}$ m$^{-3}$) at the aircraft sampling locations based on UKV data.

This air history matrix represents the mole fraction enhancement at the sample locations due to a unit flux in each grid box. By combining this information with the NAEI inventory emissions ($F_{ij}$) we can calculate a time series of simulated mole fraction enhancements ($X$) at the aircraft sample locations:

$$X(t) = \sum_{i,j} D_{ij}(t) \times F_{ij} \tag{2}$$

The emissions $F_{ij}$ are given here in units of $mol_X$ m$^{-2}$ s$^{-1}$, where X represents either $CO_2$, $CH_4$ or CO. Figures 2d, 2e and 2f show the equivalent data to Figs. 2a, 2b and 2c, coloured by simulated mole fraction enhancement rather than measured mole fraction.

## 3. Inventory flux comparisons

In this section we present two approaches to assess the accuracy of the NAEI inventory emission values relative to the measured mole fractions during this case study. The first is a new approach, referred to hereafter as the flux-dispersion method, using the simulated mole fraction enhancements from Sect. 2.3 to derive simulated fluxes through the downwind sampling plane based on inventory emissions, thus enabling comparison with corresponding measured fluxes. The results from this method represent our best assessment of inventory fluxes for this case study.

In Sect. 3.2 we then employ a conventional mass balance method to derive top-down fluxes which are compared to an aggregated NAEI value. We discuss the outcomes of both approaches in Sect. 3.3 and explain how the conventional mass balance approach can lead to spurious conclusions in cases such as this.

It is important to note that the NAEI contains only annually averaged emissions and so does not capture the potentially large temporal variability on diurnal, weekly and seasonal timescales. Clearly this represents a likely source of difference between the top-down results derived from our single flight case study (which represent a snapshot in time) and the inventory; this is discussed further in Sect. 3.1.2. The most recent gridded emissions available in the NAEI at the time of writing were for the year 2015; therefore we have used these 2015 emissions to represent the 2016 values in both approaches. The UK totals (not spatially disaggregated) for 2016 have been released, allowing us to compare these to the 2015 totals. For CO there was a 9.4% reduction in total reported emissions between 2015 and 2016, while for $CO_2$ there was a reduction of 5.8% and for $CH_4$ there was an increase of 0.01%. These inter-annual changes are likely to be small in comparison with the variability on shorter timescales mentioned above.

## 3.1 Flux-dispersion method

### 3.1.1 Methodology

To make a comparison between the measured and simulated datasets described in Sect. 2 it is first necessary to calculate a background mole fraction for both, so that the mole fraction enhancement due to the London plume can be determined. To determine periods of sampling that were not significantly influenced by the London plume, and therefore can be considered to represent background mole fractions, we again utilised the air history information given by the NAME dispersion modelling. From the gridded air histories described in Sect. 2.3, we calculated the fraction of $D_{ij}(t)$ that was within the Greater London administrative region for each release period, and defined all release periods where this fraction was less than 0.05% as background periods. This Greater London fraction is shown in Fig. 3, with the background periods coloured red.

In practice there is no sharp distinction between in-plume and background sampling, so any criteria used to separate sampling into these two categories inherently involves some level of human judgement. The key consideration when defining the background for use with this method is to use a threshold that optimises the sensitivity of the results to the region of interest, in this case Greater London. This is illustrated by Fig. 4, which shows the air history ($D_{ij}(t)$) aggregated over both the background (Fig 4a) and in-plume periods (Fig 4b). This clearly illustrates that the background criteria used here avoids air histories with significant influence from the London conurbation.

The comparison between measured and simulated flux discussed later in this section is a comparison between the flux enhancement from the areas sampled in Fig. 4b relative to the flux enhancement from the air histories sampled in Fig. 4a. Clearly this comparison is not entirely selective of emissions from Greater London, with additional influence from emissions within a wider area (largely upwind and downwind of Greater London). However, given the sampling strategy employed it is not possible to isolate Greater London emissions from other upwind and downwind sources using any technique, and the 0.05% threshold employed represents the best choice to isolate sampling periods with significant Greater London influence. While these extraneous emission sources reduce the selectivity of the results derived using the flux-

dispersion method towards Greater London, a key advantage of this technique is that these emissions do not bias the overall results, in contrast to the mass balance method. This is discussed further in Sect. 3.3.

For both the measured and simulated datasets the mole fraction enhancement due to the London plume is calculated by subtracting the background mole fraction. For each constant-altitude aircraft transect we calculated average background mole fractions to the north and south of the plume separately, for both measured and simulated datasets. We then calculated the mole fraction enhancement, $\Delta X_{London}(t)$, for both datasets using Eq. (3):

$$\Delta X_{London}(t) = X(t) - \frac{(\overline{X_{bgd\,N}}(z) + \overline{X_{bgd\,S}}(z))}{2} \tag{3}$$

Here $X(t)$ is the mole fraction time series and $\overline{X_{bgd\,N}}(z)$ and $\overline{X_{bgd\,S}}(z)$ are the average background mole fractions to the north and south of the plume for each transect. The motivation for treating the background in this way is to capture potential latitudinal and vertical gradients in mole fraction. Vertical gradients are accounted for by calculating a separate background value for each transect, while latitudinal gradients are accounted for by the separate calculation of the north and south backgrounds. If a straight average over all background periods was used for each transect, this would be subject to potential bias in cases where there was more background sampling on one side of the plume than the other. Calculating north and south backgrounds separately as in Eq. (3) mitigates this issue. An alternative method is to interpolate the background values between the north and south edges of the plume, however due to the symmetry of the plume in this case interpolating rather than averaging had a negligible impact on the results (changing the final ratios by less than 1%).

The background values used in Eq. (3) are given in Table 1. It is notable that the simulated CO and $CH_4$ mole fractions were higher to the north of the plume than to the south, but this gradient was not observed in the measured mole fractions. This could be indicative of an incorrect spatial distribution of emissions within the inventory, which would contradict one of the inherent assumptions made using this method. Alternatively, the higher simulated background to the north of the plume could be counteracted in the measured dataset if this air had a lower initial mole fraction before reaching the British Isles (as the simulated dataset implicitly assumes a uniform background for air entering the domain). The treatment of the background here can mitigate either issue as long as they result in a linear change in mole fraction with latitude. Clearly any non-linear effects have the potential to bias the final results, for example if the air that passed over London had a higher mole fraction before reaching the British Isles than both the background sampling to the north and the south of the plume. While this cannot be entirely ruled out, Fig. 4 shows that (according to NAME) the air sampled during the in-plume periods and the air sampled during the background periods had very similar entry points into the domain, reducing the likelihood that this is a significant source of bias.

The time series of measured and simulated mole fraction enhancements calculated using Eq. (3) are directly comparable quantities. However, the simulated mole fraction enhancements are strongly dependent on the model wind speeds (which have a direct impact on the average particle residence time in Eq. (1)). Any bias in the model wind speeds relative to the measured wind speeds consequently produces a bias in the simulated mole fraction enhancements. Figure 5

shows a comparison of modelled and measured wind speeds throughout the course of the flight. It can be seen that the model tends to overestimate wind speed within the boundary layer, particularly at lower altitudes.

In order to account for the low-biased simulated enhancements resulting from the high-biased model wind speeds, we convert both measured and simulated mole fraction enhancements into fluxes per unit area in the mean wind direction (i.e. through the downwind sampling plane) before making a comparison between them. To define a representative wind direction, we took the average of the mean UKV model wind direction and the mean measured wind direction during the sampling period. A time series of flux per unit area in this average wind direction, hereafter referred to as the flux density, was then calculated for both measured and simulated datasets using Eq. (4):

$$FD(t) = \Delta X_{London}(t) \times n_{air}(t) \times U_\perp(t) \tag{4}$$

The mole fraction enhancement ($\Delta X_{London}$), molar density of air ($n_{air}$) and wind speed in the mean wind direction ($U_\perp$) were calculated independently for the measured and simulated datasets, producing flux densities ($FD$) in mol m$^{-2}$ s$^{-1}$ that are directly comparable.

### 3.1.2 Flux-dispersion results

Figure 6 shows a comparison between these measured and simulated flux densities as a function of latitude for each plume transect. The lowest transect from Figs. 2 and 3 (~120 m) has been excluded because no value for $\overline{X_{bgd\,N}}$ was obtained – this was the first transect conducted before the position of the plume had been fully established so its northern extent was not sampled. The top two transects have also been excluded here because they are entirely above the average boundary layer height of 759 m used by the NAME dispersion model for this simulation (which is taken directly from the UKV met data). Above this height the parameters used by NAME to describe the turbulent motion of the particles are set to fixed values resulting in poorer representation of particle dispersion. Within the boundary layer these parameters are calculated from the friction velocity and characteristic convective velocity. Therefore the flux densities calculated for transects within the model boundary layer are more accurate than those above it.

We note that the flux density enhancements for the two transects above the model boundary layer are underestimated by the simulation. A possible cause for this would be suppressed vertical mixing in the model as a result of the simplified turbulence parameterisation above this height. A full investigation into the impact of turbulence parameterisation on the vertical mixing within the NAME model would require a separate study, but we note that if the vertical mixing in the model is suppressed this could represent a potential source of bias, leading to larger simulated flux densities within the boundary layer than would in reality be produced by the inventory emissions.

A notable feature of the transects shown in Fig. 6 is that the centre of the simulated plume is consistently further north than the centre of the measured plume. This could suggest the spatial distribution of emissions within the inventory is incorrectly weighted towards sources in the north of London. Alternatively, any inaccuracy in the model wind field could lead the simulated plume to be advected to a more northerly position on the sampling plane than the measured plume. The

fact that all three species exhibit the same northerly offset of the simulated plume points to the latter explanation, as each species has a different source mix, making it unlikely that they would all exhibit the same spatial bias.

This mismatch in plume position has the potential to impact the results, as it suggests the air histories for in-plume and background periods simulated by NAME may differ slightly from the actual air histories of the measurements. In addition, the high bias of the simulated wind speeds relative to the measurements (shown in Fig. 5) could potentially result in simulated air histories which underestimate the cross-wind spread in the sample footprint. A robust quantification of the uncertainty associated with model wind field inaccuracy would require an ensemble of NAME runs to be performed, driven by met data with perturbed wind fields. Such quantification is beyond the scope of this study, but we note that this is a potentially significant source of uncertainty in the context of the uncertainty ranges calculated below.

Having calculated time series of flux density for the measured and simulated datasets, we then calculated average flux densities for each transect altitude. We also calculated flux densities as an overall average using data from all three transects. These values are given in Table 2 for $CO_2$, $CH_4$ and CO, along with the ratios between measured and simulated flux densities. These ratios represent the factors by which the NAEI inventory needs to be multiplied in order to reproduce the measured flux densities (assuming there is no bias induced by the NAME transport). Therefore for this case study we conclude that NAEI emissions require scaling by the overall values of 1.00 for CO, 0.70 for $CH_4$ and 1.57 for $CO_2$. It is worth stating again that these values are not solely influenced by Greater London emissions, with non-negligible influence from surrounding emission sources, in particular sources immediately upwind and downwind of London. However, through our choice of background (as described in Sect. 3.1.1) we have maximised the sensitivity of the results to emissions from Greater London.

While there are small uncertainties associated with the measured mole fractions (as discussed in Sect. 2.1), the uncertainty in these overall inventory scale factors is expected to be dominated by NAME transport uncertainty. As discussed above, quantification of the uncertainties associated with the dispersion modelling would require a more involved modelling study using an ensemble of NAME runs. Here we take the range of scale factors across the different transects of 0.90-1.13 for CO, 0.61-0.79 for $CH_4$ and 1.40-1.77 for $CO_2$ to give an indication of the total scale factor uncertainty for each species, while noting that there may be additional sources of model transport bias that are not captured by this range.

When considering the implications of the scale factors derived here using the flux-dispersion method, it is important to remember that these correspond to a single flight case study. The variability in emissions on diurnal, weekly and seasonal timescales is potentially large, which places an important caveat on inferences drawn regarding the annual emissions in the NAEI. With this caveat, and considering both the central estimate and uncertainty range, we can conclude that the CO measurements taken during this case study are consistent with the emissions given in the NAEI inventory. For $CH_4$, however, the NAEI emissions yield an overestimate of flux density enhancement for each transect, and overall require downscaling by a factor of 0.70 to agree with observations. This is qualitatively consistent with studies suggesting national NAEI $CH_4$ emission totals have been too high in previous years (e.g. Manning et al., 2011; Ganesan et al., 2015) but it

differs from the most recent top-down verification report (O'Doherty et al., 2017), which finds good agreement between the NAEI $CH_4$ emission totals and continuous ground-based measurements in recent years.

Temporal variability in emissions is an obvious source of difference between our results for $CH_4$ and those in the verification report. Helfter et al. (2016) used an eddy-covariance technique to determine the diurnal variability of London $CH_4$ emissions, and found that emissions increased by a factor of 1.9 (maximum-to-minimum ratio) between the early and late morning. However, in addition to true temporal variability (e.g. rush hour emissions), such techniques are susceptible to the complex nature of urban boundary layer evolution during these transition periods (Halios and Barlow, 2018), which can result in overestimation of diurnal flux variability.

Another possible source of discrepancy with the verification report is the different sample footprint in our study. It is possible that although NAEI emission totals agree with long term observations, the spatial distribution of these emissions is not well represented in the inventory, such that the proportion of emissions ascribed to urban areas is too large. Alternatively, this difference could represent a low-bias in our results associated with inaccuracy in the wind field driving the dispersion modelling (see discussion above). To reduce the impact of both temporal emission variability and random error in the dispersion modelling, allowing for a more robust comparison with the annual NAEI emissions, repeated flights at different times of day, week and year would be required.

For $CO_2$ we find that the NAEI would require upscaling in order to be consistent with observations. However, direct comparison with the NAEI is not appropriate for $CO_2$ because biospheric fluxes, which are not included in the NAEI, represent a significant influence on the measured mole fractions. These biospheric fluxes include uptake due to photosynthesis (gross primary production; GPP), emission from autotrophic respiration and emission from heterotrophic respiration. Net primary production (NPP) is calculated as the difference between photosynthetic uptake and autotrophic respiration. Hardiman et al. (2017) investigated biospheric $CO_2$ fluxes in Massachusetts and found higher NPP values outside the Boston conurbation. Combined with higher heterotrophic respiration in more populated areas (including human respiration), these higher rural NPP values result in a positive net biospheric flux from urban centres relative to their surrounding areas. As we have not accounted for this net biospheric flux in our simulated flux densities we expect them to underestimate the measured values, even if the NAEI emissions are entirely accurate.

Prior quantification of the biospheric impact on the derived scale factor would require the use of an ecosystem model, and is beyond the scope of this study. However, some inferences can be made by considering the different scale factors derived for CO and $CO_2$, as these species share many of the same combustion sources. Previous studies (e.g. O'Doherty et al., 2013) have used CO measurements as a proxy for anthropogenic $CO_2$, relying on the assumption that the inventory ratio for CO:$CO_2$ emissions is correct. In this case, that would imply that the difference in net biospheric flux between the in-plume and background sampling amounted to over half the corresponding difference in anthropogenic flux (comparing the scale factors of 1.00 for CO and 1.57 for $CO_2$). However, while this comparison can be considered indicative of the potential order of magnitude for net biospheric flux, uncertainty in the inventory CO:$CO_2$ emission ratio limits our

ability to use this method to obtain quantitative information on biospheric fluxes (see Turnbull et al., 2006, for further discussion on the use of $CO:CO_2$ ratios for this purpose).

## 3.2 Conventional mass balance method

### 3.2.1 Methodology

Detailed descriptions of the mass balance technique in the context of measuring urban GHG emissions are provided by many sources. In general, in the context of bulk area flux measurement, these sources can be categorized into two basic approaches; either the emissions are assumed to be well mixed up to a given height at which they are capped by a temperature inversion (Turnbull et al., 2011; Karion et al., 2013; Smith et al., 2015), or the vertically varying shape of the plume is derived by interpolation between transects flown at multiple altitudes (Mays et al., 2009; Cambaliza et al., 2014;

O'Shea et al., 2014a), often using a kriging approach. Figures 2a, 2b and 2c clearly show that the assumptions of the first of these approaches (i.e. well mixed plumes up to a capping height) are not met in this case. We therefore adopt the latter of these approaches, and use kriging to represent the full structure of the plume. This approach necessarily assumes temporal invariance of the plume over the period of sampling: in this case ~2.5 hours.

Following the work of (Mays et al., 2009; Cambaliza et al., 2014; O'Shea et al., 2014a) we derive fluxes using Eq.

15  (5):

$$F = \int_0^{z_{max}} \int_A^B \left(X_{ij} - X_0\right) n_{air}(z) \, U_{\perp ij} \, dx \, dz \tag{5}$$

Here $F$ (mol s$^{-1}$) is the bulk flux for the emission source, $X_{ij}$ is the kriged mole fraction for a given species, $X_0$ is the background mole fraction, $n_{air}(z)$ is the molar air density (here derived as a linear function of altitude based on measured values) and $U_{\perp ij}$ is the kriged wind speed perpendicular to the vertical sample plane across which the integral is

taken.

Kriging is an interpolation method based on a stochastic Gaussian model, and is described in detail by Kitanidis (1997). It converts samples with sparse spatial coverage into a 2D grid of estimated values, with an associated grid of standard errors for these values. Here we use a modified version of the EasyKrig software (© Dezhang Chu and Woods Hole Ocean Institution) to perform the kriging; again more detail regarding the application of this software with regards to aircraft

mass balance flux calculations is given by Mays et al. (2009). More detail regarding the kriging parameters used is included in the supplementary material.

The results from the kriging were output on a 20 x 29 cell grid, with a vertical resolution of 50 m and a horizontal resolution of 5 km respectively, as shown in Fig. 7. As the lowest transect was conducted at ~120 m altitude, the structure of the plume below this level was not constrained well by our sampling. Therefore the mole fractions for the lowest 100 m

above ground level were taken to be the same as the kriged output for the layer at 100 – 150 m.

The background mole fraction $X_0$ should be chosen to best represent the mole fraction that would be measured downwind of Greater London if there were no emissions within Greater London. We determined this background for each

trace gas by taking the average mole fraction over all cells within 15 km of the north or south boundary of the sample plane (i.e. the three columns at each edge of the plane in Fig. 7). This approach follows Mays et al. (2009) in determining the background from measurements in the downwind plane outside of the influence of the plume, and contrasts with the approach used by O'Shea et al. (2014), who instead used measurements upwind of London to determine the background.

There are advantages and disadvantages to both methods. Using an upwind background approach such as in O'Shea et al. (2014), one is able to account for sources directly upwind of London in the background measurements, and avoid the influence of extraneous emissions to the north and south of London. However, either multiple transects upwind must be performed to capture the vertical mixing of upwind sources (and hence any vertical gradient in the background) or an assumption must be made that any upwind sources are well-mixed throughout the boundary layer.

A potentially more significant issue with sampling an upwind background prior to the downwind measurements is that, for a morning flight, the boundary layer height can increase significantly during the intervening time. This increase in boundary layer height is associated with entrainment of air from above the boundary layer, which consequently changes the background boundary layer mole fraction. Additionally, in this study we did not sample the same air mass upwind and downwind of London (i.e. we cannot consider this sampling to be in the Lagrangian frame), as the NAME run showed the air

took ~9 hours to cover this distance. Therefore the air history for sampling upwind of London may differ significantly from the air history for sampling downwind of London.

These issues with using upwind measurements to calculate a background mole fraction, and the lack of intense sampling within the boundary layer upwind of London in this study, makes a downwind background more appropriate in this case.  However, the influence of emission sources to the north and south of London on our calculated background, and the

failure to account for emission sources directly upwind and downwind of London, represent key sources of bias in the derived mass balance fluxes below. The impact of this issue is discussed further in Sect. 3.3.

The background mole fractions used were 147.3 ppb for CO, 1941.6 ppb for $CH_4$ and 409.1 ppm for $CO_2$, which (despite the difference in background definition) are similar to the values used in the flux-dispersion method (see Table 1). Although we have used these average background values in our main analysis, we have also calculated fluxes using

interpolated background values (as recommended by Heimburger et al., 2017) to test the sensitivity of the results to this choice of approach.

### 3.2.2 Mass balance results

The fluxes calculated using Eq. (5) are given in Table 3, along with 1σ uncertainties derived by combining the kriging standard errors with the uncertainty in background mole fraction, taken to be the standard deviation for all background cells

used. Also given are the aggregated NAEI emissions for the Greater London administrative area for each species. We have derived inventory scale factors, in principle analogous to those in Sect. 3.1.2, by taking the ratio of these aggregated NAEI emissions to the flux calculated using Eq. (5) for each species. Using the conventional mass balance method we calculate that the NAEI requires rescaling by factors of 2.27 for CO, 1.22 for $CH_4$ and 3.08 for $CO_2$. The differences between these

values and those derived in Sect. 3.1.2 are discussed in Sect. 3.3 below. Using interpolated (as opposed to average) background mole fractions slightly increases the calculated fluxes, but in all cases the difference is less than 7%. The NAEI scale factors derived using an interpolated background are calculated to be 2.33 for CO, 1.31 for $CH_4$ and 3.19 for $CO_2$.

### 3.3 Comparing the flux-dispersion and mass balance methods

In Sects. 3.1 and 3.2 two different methods were applied to the same dataset to derive scale factors for the NAEI inventory such that it agrees with aircraft observations. However, the scale factors derived using the flux-dispersion method are significantly lower than those derived using the conventional mass balance method. This is because one of the key elements of the mass balance method, the assumption that a city acts as an isolated emission source surrounded by areas with negligible emissions, is clearly violated in this case. The impact of this is discussed in Sect. 3.2.1 above; the calculated

background mole fractions are influenced by extraneous emission sources to the north and south of Greater London, but do not account for sources upwind and downwind of the conurbation. Therefore, the measured mole fraction enhancements above this background do not emanate solely from emissions within Greater London: they are representative of the difference between emissions within the air history of the in-plume measurements and emissions within the air history of the background measurements.

15          By choosing to aggregate only NAEI emissions within the Greater London administrative area we have ignored this influence of surrounding emission sources on the calculated mole fraction enhancements, but in the case of London there are significant sources of all three gases outside this administrative area. In particular, leaving upwind and downwind sources out of the inventory total biases the derived inventory scale factors high for the mass balance method. The influence of these surrounding emission sources could also explain the large inventory upscaling factors derived by O'Shea et al. (2014), who

used a mass balance approach to estimate London emissions for a previous case study in 2012. Although an upwind background was used in that study, mitigating the impact of emission sources upwind of London on the derived fluxes, this approach is still susceptible to biases associated with emissions downwind and to the north and south of London, which all contribute to the measured mole fraction enhancements in that study.

            Clearly one could choose a different region over which to aggregate NAEI emissions, but for a non-isolated source

such a choice is inherently arbitrary and yet has a direct impact on the derived inventory scale factor. Just as ignoring sources which contribute to the plume resulted here in high-biased scale factors, aggregating over too wide an area would introduce a low bias, as sources which actually contribute predominantly to the measured background are instead assumed to contribute to the plume. Emissions from many source areas contribute to some extent to both the in-plume and background measurements, making it unclear whether or not to include these in the aggregated inventory total.

30          It is worth noting that Lagrangian mass balance techniques (e.g. O'Shea et al., 2014b) and Integrative Mass Boundary Layer techniques (e.g. Font et al., 2015) frequently use dispersion model air histories to define the measured flux footprint. However, the methodology employed by these techniques, i.e. balancing the change in species concentration within a column of air against the various fluxes per unit area into and out of the column, lends itself more naturally to such

a flux footprint definition, as it does not rely on defining a spatially separate period of background sampling. Using the downwind mass balance approach here, we cannot simply attribute the derived mass balance flux to the area given by the NAME air history for in-plume sampling, as the emissions from much of this area contribute to the background sampling as well (this can clearly be seen from Fig. 4). The techniques referred to above provide alternative methods for quantifying

emissions from urban areas, although they generally have to assume emissions are well-mixed throughout the boundary layer (which is not always the case over urban areas), and obtaining permission to fly at low level directly over a city can present an additional logistical issue.

In summary, the impact of the arbitrary choice of inventory aggregation area on the conclusions drawn using the mass balance method demonstrates the inappropriateness of the downwind mass balance technique for non-isolated emission

sources. Instead, the flux-dispersion method provides a good alternative in these cases, as it is not subject to such biases.

## 5. Conclusions

Aircraft mass balance techniques are an effective way of determining emissions from isolated sources, but they require surrounding areas to be negligible emission sources in order to yield robust results. This is a well-known assumption associated with these methods. However in the absence of alternative techniques using the same sample dataset against

which the mass balance results can be compared, one is forced either to simply state this assumption as a caveat or to abandon the effort entirely.

In this study we have developed an alternative technique using a Lagrangian dispersion model to quantify the transport of inventory emissions to the aircraft sample locations, so that a direct comparison of flux per unit area can be made at the measurement locations. In contrast to the conventional mass balance technique, this method does not require

cities to be isolated from surrounding emission sources, rendering it more appropriate in many cases. We have demonstrated this new technique by applying it to a single-flight case study measuring London emissions, which yielded inventory scale factors of 1.00 (0.90-1.13) for CO, 0.70 (0.61-0.79) for $CH_4$ and 1.57 (1.40-1.77) for $CO_2$. These values represent the factors by which the inventory emissions need to be multiplied to agree with the aircraft measurements, although the absence of biospheric fluxes in the inventory means direct comparison with the $CO_2$ measurements is not appropriate. Using a mass

balance approach we derived significantly higher values (2.27, 1.22 and 3.08 respectively), which we conclude are biased as a consequence of significant sources outside the Greater London administrative region, which are neither adequately captured by the background mole fraction calculation or easy to account for in the choice of inventory aggregation area. The magnitude of this bias demonstrates how employing a mass balance method for a non-isolated source can lead to highly misleading conclusions regarding the accuracy of the emissions inventory under study.

It is important to emphasise that the inventory scale factors derived here represent the results from a single case study and therefore are not necessarily representative of the annual timescale of the NAEI emissions. In order to better validate the inventory on this timescale repeated flights following a similar sampling strategy are required. The limited

spatial selectivity of the flux-dispersion technique represents another caveat on the results from a single flight, as the derived flux ratios are not only sensitive to emissions from the London conurbation but also to emissions from a fairly wide area surrounding it. Repeated flights should therefore be designed to incorporate sampling under different prevailing wind directions, so that the systematic impact of extraneous sources on the overall results is minimised. Using the flux-dispersion

method developed here in combination with representative aircraft sampling on an annual timescale could provide a robust assessment of inventory fluxes at the city scale in the case of non-isolated sources for which the mass balance technique is not appropriate.

**Author contributions**

Conceptualization, J. P., G. A., J. L.; Data curation; J. P., S. B.; Formal analysis; J. P.; Funding acquisition; G. A., M. G., J.

L., A. M., P. P.; Investigation; J. P., G. A., S. B., J. L., W. D., B. N.; Methodology; J. P., G. A.; Software, A. M.; Supervision, G. A., M. G., J. L., P. P.; Writing – original draft, J. P.; Writing – review & editing, J. P. G. A., P. P.

**Acknowledgements**

The authors would like to thank everyone at FAAM, Airtask and Avalon Aero who helped with the collection and processing of the aircraft data used here. We acknowledge use of the NAME atmospheric dispersion model and associated NWP

meteorological data sets made available to us by the Met Office. We acknowledge the significant storage resources and analysis facilities made available to us on JASMIN by STFC CEDA along with the corresponding support teams. J. R. Pitt received a NERC CASE studentship in partnership with FAAM, grant number NE/L501/591/1, supervised by G. Allen. This work was supported by the GAUGE (Greenhouse gAs UK and Global Emissions) NERC project, grant numbers NE/K002449/1 and NE/K00221X/1.

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

| Dataset | Alt (m) | CO (ppb) | | CH$_4$ (ppb) | | CO$_2$ (ppm) | |
|---|---|---|---|---|---|---|---|
| | | South | North | South | North | South | North |
| | 287 | 148.4 | 147.4 | 1943.1 | 1941.8 | 409.6 | 409.3 |
| Measured | 460 | 149.2 | 145.3 | 1942.5 | 1938.4 | 409.3 | 408.9 |
| | 575 | 150.2 | 149.9 | 1943.6 | 1943.3 | 409.2 | 409.3 |
| | 287 | 3.7 | 4.5 | 6.8 | 11.7 | 0.5 | 0.6 |
| Simulated | 460 | 3.7 | 3.8 | 7.0 | 10.7 | 0.5 | 0.5 |
| | 575 | 3.7 | 4.3 | 6.9 | 12.7 | 0.5 | 0.5 |

**Table 1: Background mole fractions for each species to the north and the south of the London plume, calculated using the flux-dispersion method.**

| | CO | | | CH$_4$ | | | CO$_2$ | | |
| | Mean flux density ($\mu$mol m$^{-2}$ s$^{-1}$) | | | Mean flux density ($\mu$mol m$^{-2}$ s$^{-1}$) | | | Mean flux density ($\mu$mol m$^{-2}$ s$^{-1}$) | | |
| Alt (m) | Measured | Simulated | Ratio | Measured | Simulated | Ratio | Measured | Simulated | Ratio |
|---|---|---|---|---|---|---|---|---|---|
| 287 | 1.80 | 1.80 | 1.00 | 2.73 | 3.46 | 0.79 | 541 | 348 | 1.56 |
| 460 | 1.65 | 1.84 | 0.90 | 2.12 | 3.50 | 0.61 | 468 | 334 | 1.40 |
| 575 | 1.94 | 1.71 | 1.13 | 1.96 | 2.85 | 0.69 | 556 | 314 | 1.77 |
| **Overall** | 1.79 | 1.79 | **1.00** | 2.28 | 3.28 | **0.70** | 521 | 332 | **1.57** |

**Table 2: Mean flux densities calculated using the flux-dispersion method, given for each transect and taken over all three transects. The ratios between measured and simulated flux densities are all given.**

| | CO | | CH$_4$ | | CO$_2$ | |
|---|---|---|---|---|---|---|
| | Mean | 1$\sigma$ | Mean | 1$\sigma$ | Mean | 1$\sigma$ |
| Flux (kmol s$^{-1}$) | 0.178 | 0.006 | 0.182 | 0.009 | 44.7 | 1.2 |
| NAEI emissions (kmol s$^{-1}$) | 0.079 | N/A | 0.149 | N/A | 14.5 | N/A |
| **Ratio** | **2.27** | 0.07 | **1.22** | 0.06 | **3.08** | 0.08 |

**Table 3: Bulk fluxes calculated using a conventional mass balance technique and corresponding NAEI emissions, aggregated over the Greater London administrative region. The ratio of mass balance flux to NAEI emission is also given.**

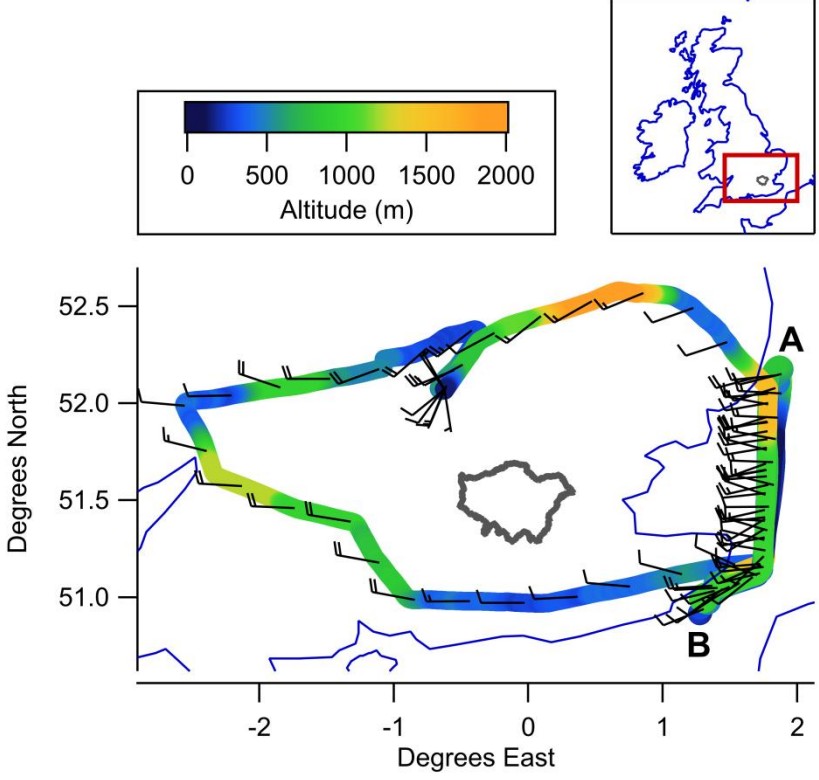

**Figure 1: Aircraft flight track on 04 March 2016, coloured by altitude. Wind barbs are used to represent wind speed and direction, averaged over 5 minutes, using the convention where each full wind barb represents a wind speed of 10 knots. The border of the Greater London administrative region is shown in grey for reference.**

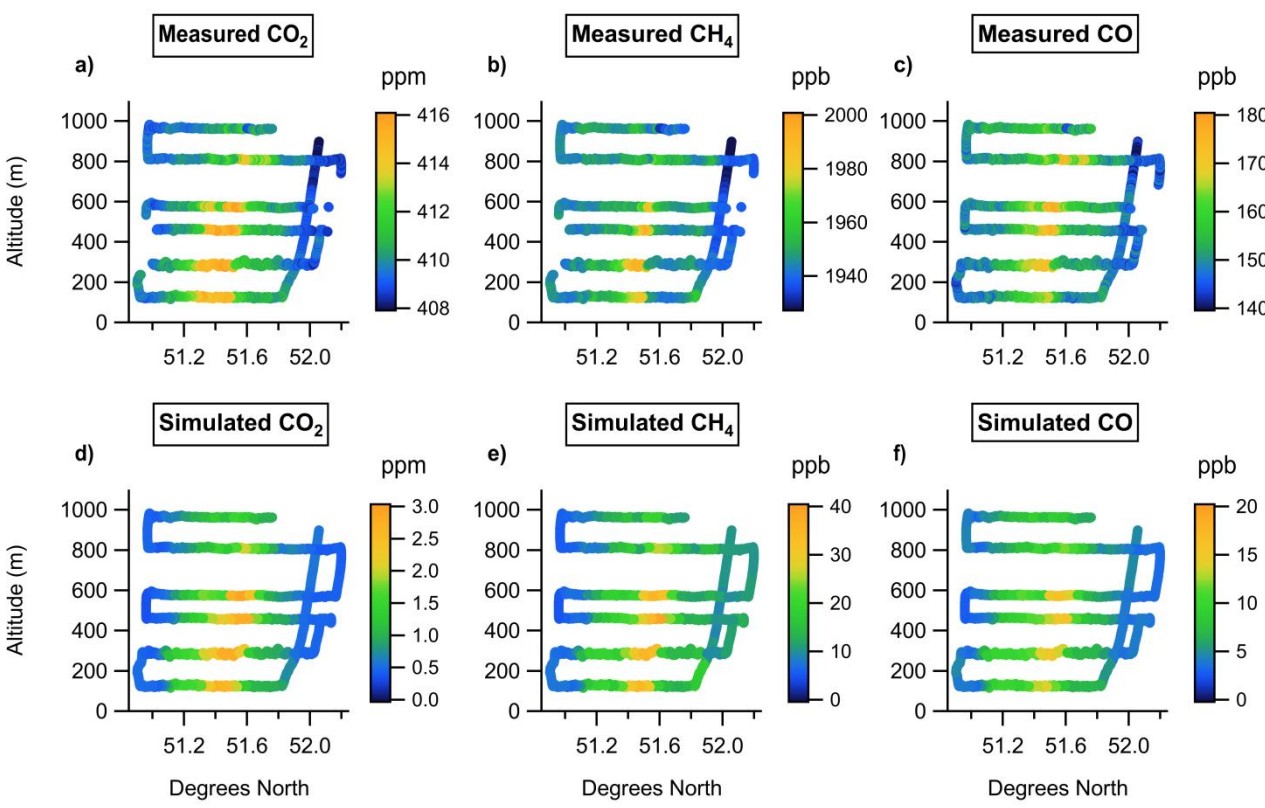

**Figure 2: Altitude-latitude projections of measured mole fraction (a – c) and simulated (d – f) mole fraction enhancement downwind of London for each species.**

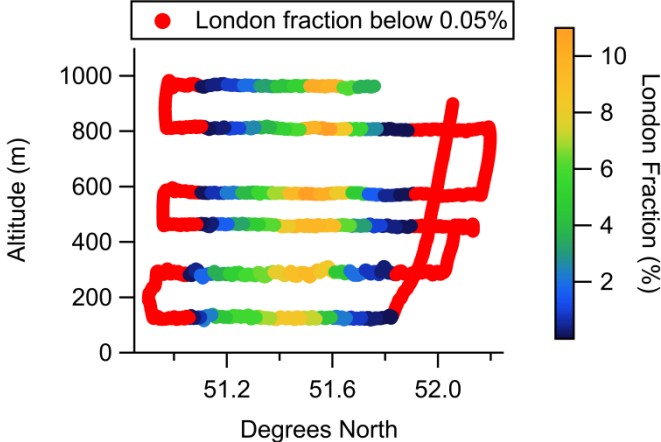

**Figure 3: Altitude-latitude projection showing the influence of London on the downwind sampling, as determined from the NAME air histories. The colour scale represents the fraction of aggregated NAME air history $D_{ij}$ within the Greater London administrative region for each NAME release period. Background periods, where the London fraction is less than 0.05%, are shown in red.**

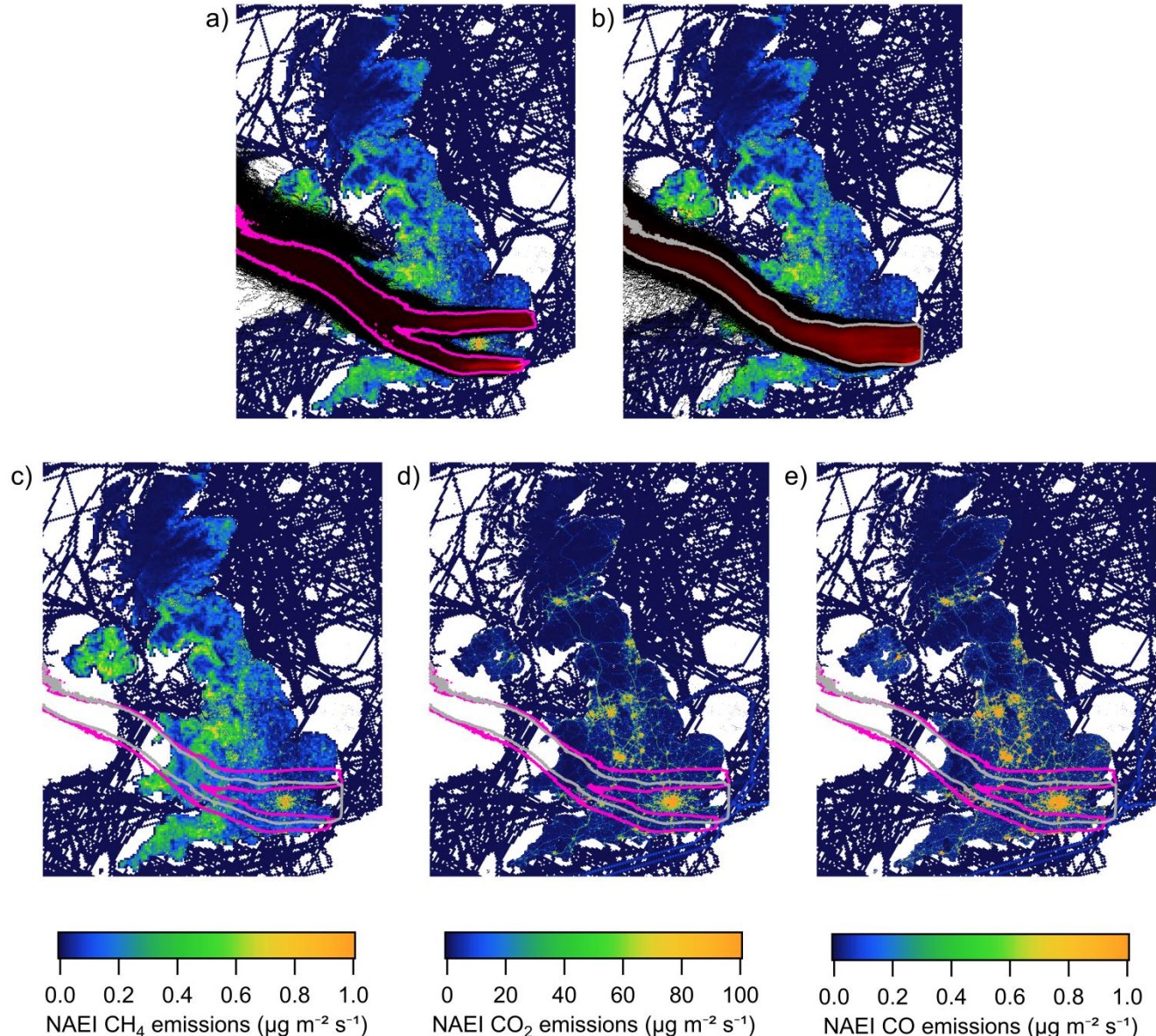

**Figure 4: NAME air histories aggregated over (a) background sampling periods, and b) in-plume sampling periods, overlaid on an NAEI emissions inventory map for CH₄ (shown using a saturated colour scale). Both air histories have been normalised such that they sum to 1, with grey and pink contours shown in each plot surrounding the vast majority (99.9995%) of sample influence. These contours are included in panel c), d) and e) to provide a better visual comparison between the two aggregate air histories in the context of the inventory emissions for CH₄, CO₂ and CO respectively.**

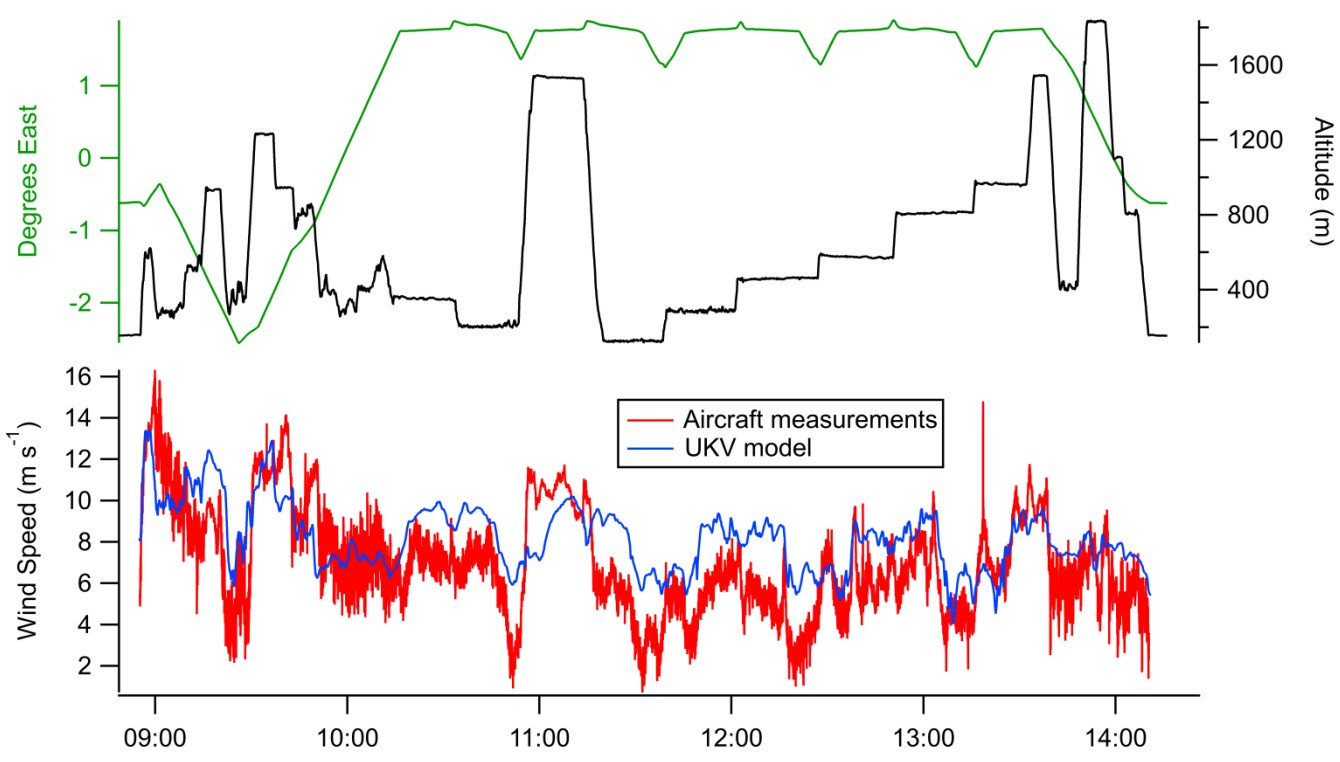

**Figure 5: Comparison between wind speeds measured by the aircraft and the corresponding wind speeds at the aircraft location from the UKV model. It can be seen that the model generally overestimates wind speed within the boundary layer.**

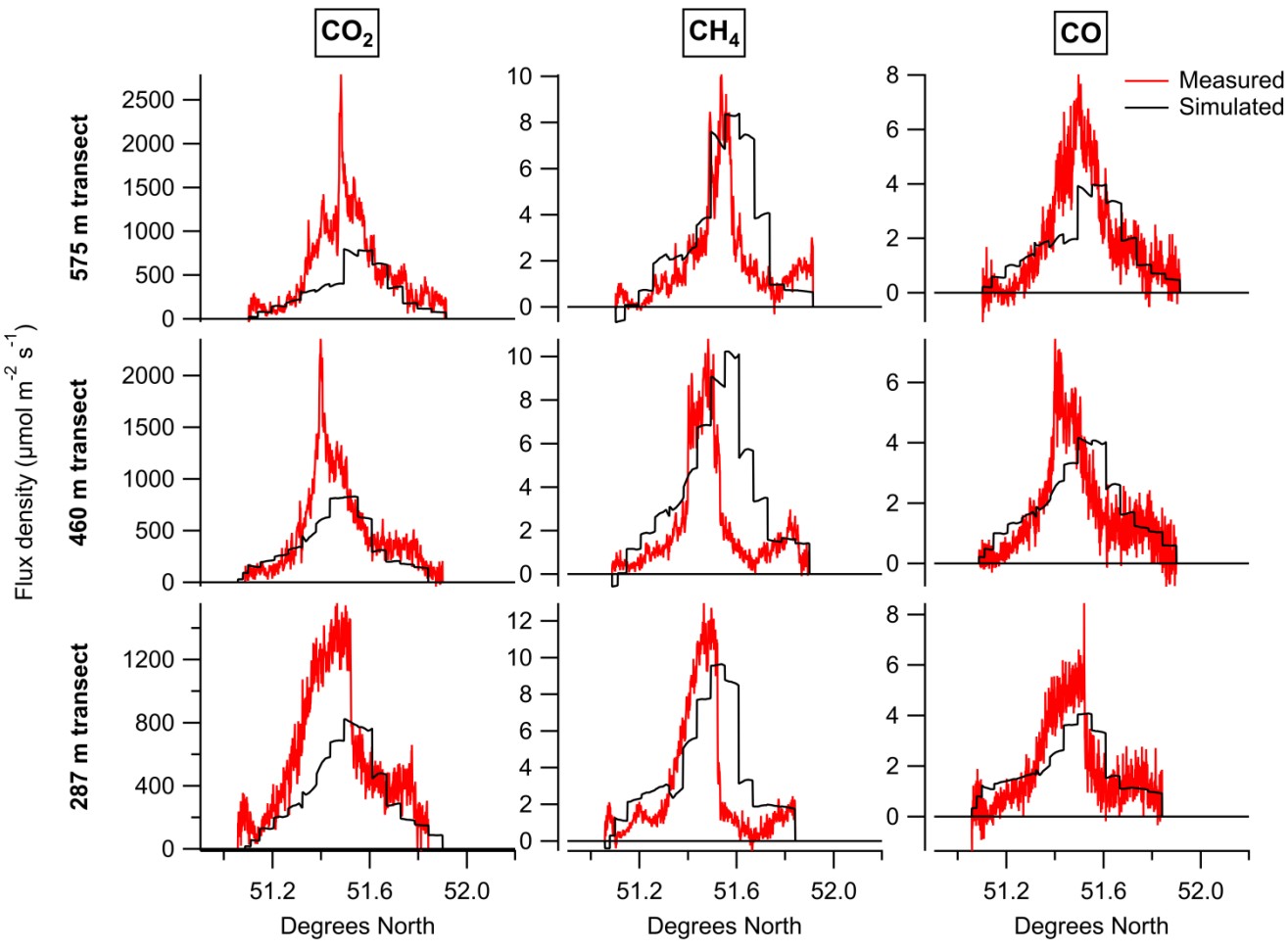

**Figure 6: Measured and simulated flux densities for $CO_2$, $CH_4$ and CO, given for the three transects (287 m, 460 m and 575 m) used to assess inventory accuracy.**

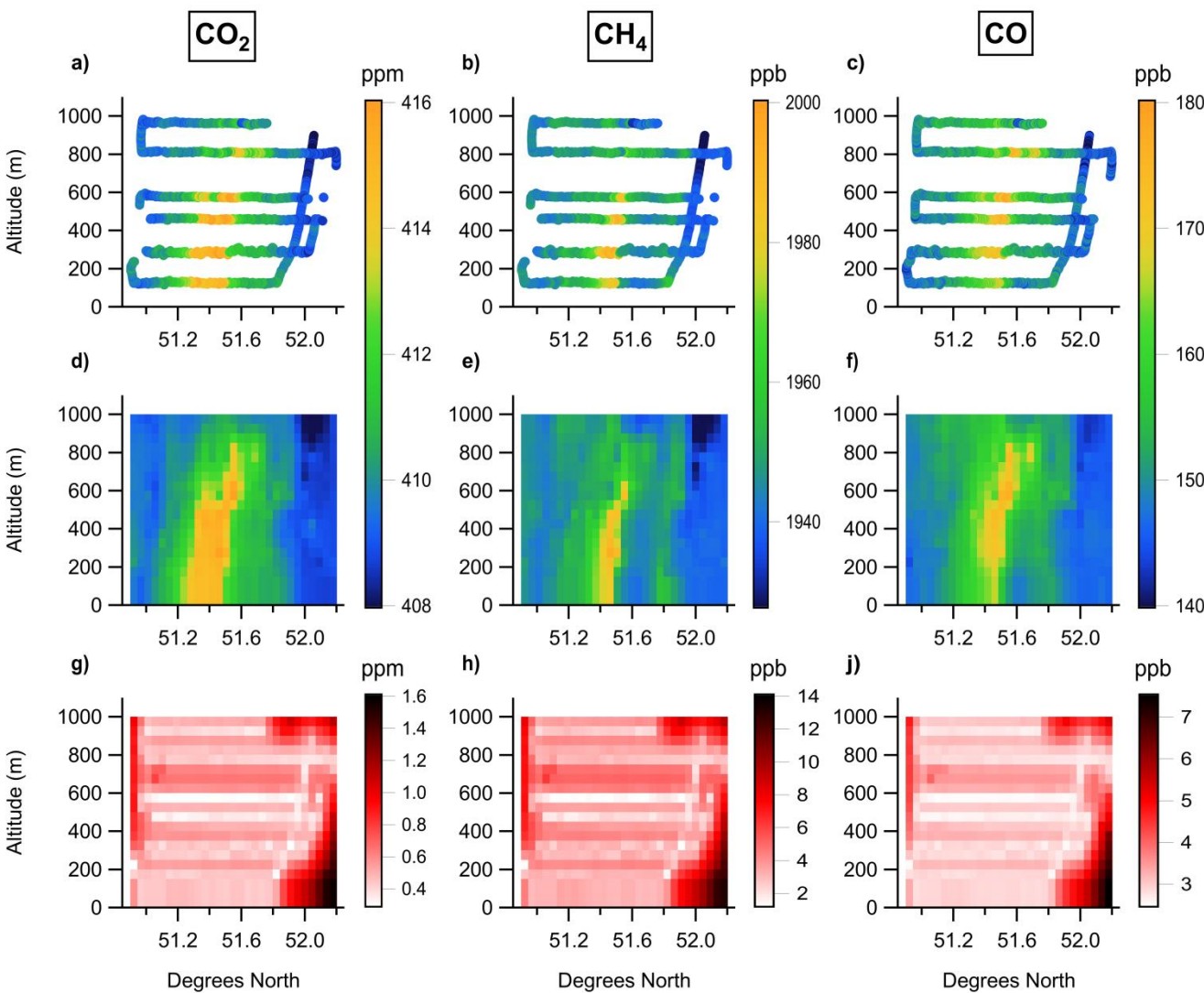

**Figure 7: Altitude-latitude projections of: a) – c) measured data, d) – f) kriged data, g) – j) kriging standard error, for CO$_2$, CH$_4$ and CO respectively.**

