# Peer review of "Assessing London CO2, CH4 and CO emissions using aircraft measurements and dispersion modelling"

_Atmospheric Chemistry and Physics, 2018_

## Referee Comment (RC1) · J. Turnbull (Referee) · 18 Nov 2018

This paper presents results of a single flight around London, comparing two techniques for evaluating emissions from the aircraft measurements. First, the authors apply Lagrangian modelling methodology to simulate the expected CO2, CH4 and CO mole fractions at each aircraft sampling location based on the UK NAEI inventory. The simulated values are compared with observations to evaluate the accuracy of the NAEI for each species. Second, the authors use the well-established mass balance method. They compare the two methods and argue that the Lagrangian modelling technique is more appropriate, particularly for a location like London where emissions outside of the

metropolitan region are not negligible.

Thank you to the authors for the excellent writing, which makes it easy for the reviewers to focus on the science. Overall, this is a nice study and it presents a straightforward methodology that will likely be used by many others in the future. I do have some concerns about the details of the comparisons, and particularly in the choice of background. Overall, the paper needs only minor revisions, noted below.

Specific comments: Pg 1 ln 19-20. Consider rephrasing – many in the atmospheric greenhouse gas community are recognizing that the value of atmospheric measurements in the emissions reporting context is in working with existing inventories to evaluate and improve emissions reporting. Presenting these measurements as "independent verification" pitted against inventory methods is problematic.

Pg 2 ln 3-10. Please add some discussion here about sources/sinks that are not included in the NAEI. For CO2 this is mainly biogenic fluxes, which are noted later to be critically important. Please also include a discussion of what sources of CO and CH4 are included in the inventory, and which are not. For example, I suspect that oxidation of biogenic VOCs is not included in the CO inventory. These may be negligible in March, but should still be mentioned. As for the CH4 inventory, does it include all sources, or only anthropogenic sources, and how significant might non-anthropogenic sources be?

Pg 2 lines 21-22. While it is true that comparison of top-down estimates with bottom-up inventories is one important way to use the atmospheric observations, it is certainly not true that the only use of these measurements is to evaluate inventories! Please rephrase.

Pg 4 ln 25. "an altitude-latitude plane."

Pg 5 ln 6-20. Please add a sentence that explains in plain English the principle of what the equations do, rather than requiring the reader to wade through the equations to

figure out the principle (although the detail of the equations is necessary too).

Pg 5 section 3. Please add some detail about the NAEI. It is spatially explicit, but does it have temporal variability? If so, what kind of temporal variability and how reliable might that be? Diurnal cycles? Seasonal cycles? Weekday/weekend? Are there any existing estimates of the quality of the inventory (and perhaps the quality is different for the different gases)? This becomes important in trying to understand the differences between the inventory and the observations.

Pg 6 ln 24-25. Again, please add a sentence that explains the principle in plain English rather than forcing the reader to work it out from the equations. Eg "The mole fraction enhancement is calculated by subtracting the background value".

Pg 6 ln 24-32. The choice of background is known to be a key uncertainty in this type of measurement (eg. Cambaliza et al 2013; Heimburger et al, 2017). Unfortunately the research community has not yet come to any conclusion as to how to resolve this. The simple method of taking an average of the values measured on the downwind edges of the plume (as is done here) is far from perfect, even if it might be the best available option given the measurements that have been done. Heimberger et al (2017) showed that there can be significant differences in the values on the two edges, and that in that case, a simple improvement would be to linearly interpolate between the two edges to evaluate background. It is also entirely possible that the background is not uniform and that there are plumes from upwind sources that are not detected because they are inside the urban plume. From Figure 4, it's apparent that there are a lot of methane emissions upwind of the city that could cause this. Further, there's an implicit assumption that there are no emissions occurring in the footprint of the edge measurements. This is clearly a bad assumption for this dataset, and so the edge measurements will be biased high (or perhaps low in the case of CO2 if there is significant drawdown in the edges), resulting in an underestimate of the urban emission rate (or perhaps overestimate in the case of CO2). A forthcoming paper (in last phases of review) will discuss this further, but unfortunately is unlikely to be published in time

to be referenced in this paper. My suggestion is to: (1) Add a figure that shows clearly the background values, how they were chosen, and whether there is any difference between the two edges. (2) A plot of the upwind measurements could also be included to show whether there is any particular concern with plumes coming in from upwind for this dataset. (3) Add figures that show the NAEI $CO_2$ and CO emissions, similar to that shown in Figure 4 for $CH_4$, to give a sense of upwind and edge emissions and how important they might be. (4) If there are no particular concerns with the points above, then stick with the current choice of background. (5) Add some discussion about the uncertainty associated with choice of background and how it might influence the results.

Pg 7 ln 24-30. Looking at figure 5, there's a clear spatial mismatch in the plume location between the obs and simulation. What might be the explanation for this? Given this mismatch, is it reasonable to average over the whole thing and then compare the two methods? This mismatch seems to imply a larger uncertainty than that given by just comparing the means.

Pg 8. Please emphasize throughout the discussion of the comparison that this analysis is for a single flight, and that care should be taken in drawing conclusions about the integrity of the inventory from a single comparison on a single day. Previous authors have shown that when multiple flights are considered, there can be large differences in the calculated flux that are likely due to uncertainties in the top-down flux estimate rather than day-to-day differences in the actual emissions.

Pg 8 ln 10-23. I agree that an incorrect spatial pattern in the inventory could explain at least part of the difference. However, I suspect that the choice of background may be more important and be biasing the top-down estimate low. See earlier comments. Does the NAEI include temporal variability and could lack of temporal variability in the NAEI be an explanation for the difference? See earlier comment.

Pg 8 ln 24 – 35. It's clear than biogenic $CO_2$ will have an enormous influence on the

calculated flux, and that it can bias the CO2 background quite dramatically (see e.g. Turnbull et al 2015, Cambaliza et al 2013). The statement here needs to be much stronger, "treated with caution" is an understatement! It is simply not possible to compare a flux based on total CO2 with an anthropogenic CO2 inventory unless the biogenic component can be accounted for, likely by either having a good biogenic model or being able to separate biogenic and fossil fuel CO2 in the observations (e.g. using 14C or CO). I would say something like "comparison with the NAEI is not appropriate for this dataset".

Pg 9 ln 28-34. See earlier comments about choice of background. The same biases occur for this method as for the other method.

Pg 10 Section 3.2.2. Can you come up with a total emission flux for the flux dispersion method, so that the total flux from each method can be compared more directly? As written, the comparison is between the obs/model ratio for each of the two methods. Thus it can't be determined whether the difference in the ratio occurs because the observed flux rate is different, or the modeled flux rate is different. You argue that the difference is in the modeled flux (actually that you've defined the modelled footprint differently in the two cases). By making a slightly different comparison, this could be argued more strongly.

Pg 10 section 3.3. This difference in how the footprint is defined is a good candidate for the difference. There are potentially ways to resolve this in the mass balance method. A good start would be to make an estimate of the footprint of the mass balance, rather than assuming that the footprint is an arbitrary metropolitan boundary.

Pg 11. Conclusions. Please restate the point that the CO2 comparison is invalid because biogenic CO2 is not accounted for. Otherwise the conclusions are very nice.

Figure 1. Please add a larger scale inset to show where this is in relation to the UK, Ireland, etc. Not all of us are well-versed in English geography!

[Figure]

---

## Referee Comment (RC2) · Anonymous Referee #2 · 27 Nov 2018

The manuscript "Assessing London CO2, CH4 and CO emissions using aircraft measurements and dispersion modelling" by Pitt et al. uses an aircraft atmospheric measurement campaign to estimate the CO2, CH4 and CO emissions from the Greater London area. They propose a new approach to estimate these emissions from the airborne concentration and wind measurements and they compare the results from this new approach to that from the more traditional mass balance technique. Their results seem to indicate that the mass balance approach suffers from the lack of knowledge on the footprint of the corresponding flux computation when the targeted source is not isolated. In contrast, their new approach takes advantage of atmospheric transport simulations to better connect the fluxes computed along the aircraft transects to the

surface emissions.

First, I would like to mention that I really appreciated the concision, clarity and quality of the text. The study is based on a well planned measurement campaign and on robust principles of computation. The analysis of the two estimation methods is interesting. This study is clearly worth being published. However, I would encourage to deepen the discussions and maybe analysis in order to better characterize the concepts, strengths and weaknesses of the methods.

1) My understanding is that the new approach is merely a combination or trade-off between two traditional approaches : the mass balance approach and the atmospheric transport inversion (Brioude et al. 2013 provide an example of inversion applied to aircraft data around a city). Conceptually, the major difference between this approach and the traditional atmospheric transport inversion is related to the fact that the observed variables to be fitted by rescaling the surface fluxes are fluxes at the measurement locations rather than concentrations. This requires some additional assumptions for the computation of such fluxes, but this enables to account for wind measurements when assimilating the observations. Another difference is that rather than assimilating all local fluxes at the aircraft measurement locations in a Bayesian statistical inversion framework, the method consists here in summarizing them into an average value which is used to rescale the map of surface fluxes. This simplification could lead to a loss of information but it can also help control the inversion behavior. One of the strength of traditional atmospheric inversions and of this new approach is the ability to extrapolate the information from the sparse measurements by accounting for the atmospheric transport and for the emissions spatial distribution, while the traditional mass balance approach makes coarser extrapolations (here based on a kriging technique).

I think that such a comparison to the atmospheric inversion is worth being discussed since the comparison to the mass balance approach only could lack of hindsight regarding the panel of methods that have been tested to exploit aircraft data. Furthermore, from my point of view, this new approach is closer to the atmospheric transport

inversion than to the mass balance approach.

2) One of my main concerns is that by rescaling the total of the NAEI emissions according to measurements whose surface footprint extends well beyond the Greater London area, the new approach does not really inform on the emissions from this area either. Given the distances from the section A-B to London, and as illustrated by Figure 4, the results from this method are driven by emissions from a large part of the South of England that extends to the sea, despite the removal of the "background" concentrations (whose sensitivity to the Western part of the South of England seems much smaller than that of the measurements used to constrain the estimate of emissions according to Figure 4).

The computations are conducted in March so that ignoring the natural $CO_2$ fluxes might be fine. But similar computations based on the same aircraft campaigns in spring and summer would be highly hampered by the $CO_2$ uptake upwind and downwind London (not only by the differences between the natural fluxes within the urban part of the measurement footprint vs. within the background footprint that are discussed in section 2.3). While the lack of account for natural $CO_2$ fluxes is mentioned in section 3.1.2, the major issues raised by these fluxes for spring / summer deserve a discussion, and the topic could deserve some indications in the method sections (in particular in section 2.3) and maybe a coarse look at estimates of the $CO_2$ natural fluxes in the UK.

I feel that the manuscript is a bit severe with the mass balance approach by crudely attributing the flux estimate from this method to the Greater London area, and maybe by deriving an estimate of the background concentrations for this approach in a crude way. More cautious interpretations of the flux estimates from this approach are usually made, especially for situations like that of London. I would recommend the authors to comment on the paper by Font et al. (2015) who also made estimates of the emissions from London using aircraft data, and who used FLEXPART simulations to assess the footprint of their measurements. O'Shea et al. (2014) also used NAME to analyze the footprint of their aircraft measurements, and discussed the issue that would be raised

by the crude assumption that these measurements would correspond exactly to the greater London area.

Therefore, I would be ready to agree that the mass balance approach and its associated type of aircraft measurement tracks is not very well adapted to the monitoring of the emissions from a city surrounded by other cities and productive ecosystems, especially if flight regulations impose measurements to be conducted far downwind. However, I feel that by relying on the same type of measurements and by avoiding to solve for the spatial distribution of the emissions, the new approach may bear the same fundamental limitation which is the lack of ability for isolating the budget of the emissions from the targeted city. In this regard, I think that the conclusions are a bit optimistic.

3) A critical variable in the study is the wind which is used to compute fluxes. Comparisons between measured and modeled (UK Met Office) winds along the transects but also all around the London Greater area could potentially provide some strong insights on the robustness of the transport model, of the estimate of the measurements spatiotemporal footprint and of the estimate of the surface emissions (in particular if biases arise in the comparisons). I feel that it deserves some analysis.

More detailed comments:

* Introduction

- p2l1: explain that "top-down" relates to methods based on atmospheric measurements and models ?

- p2l5-6: do power plant represent a large fraction of the $CO_2$ emissions in the greater London area ? on the same topic: I had in mind that the city had large power plants in its vicinity that could represent a major share of the emissions in the measurement footprint (http://naei.beis.gov.uk/data/gis-mapping): is it the case ? if yes, it would feed my main concern (2).

- p2l15: I am not sure about the meaning of "bulk area flux" here. What would prevent atmospheric inversion to provide such a bulk area flux based on the same data ? see my main point (1)

- p2l21-33: I feel that the problem of defining the footprint of the estimated flux is presented in an "inverted" way which makes things more complicated than they are. In particular there is no reason to necessarily involve inventories in this problem.

* section 2.2

- this section should provide the duration and the period of the day corresponding to the flight. Maybe I missed it in the following, but the time of the measurements is a critical information that can raise questions regarding the temporal representativity and the robustness of the computations

- p4l11: I do not understand the end of the sentence ("so as to assess the representativeness . . .") in its context

- p4l13: "less than 24 hours" -> 24 hours is large if considering the need to connect the measurements to an emission footprint both in space and in time, and given the strong diurnal variations of the fluxes. Can the statement be more precise based on NAME simulations ?

* section 3 (beginning)

-p5l30: maybe you should already clarify here the fact that NAEI provides annual budgets of the emissions only, while the measurement were made during daytime in March, which corresponds to a period of relatively high emissions (this information is limited to the discussions on the CH4 results, and just ignored for CO and CO2 in section 3.1.2). Using constant emissions in the model may also be problematic because the duration of the measurement campaign is about 2.5h, during a period of the day when emissions could be highly variable.

-p6l1-3: this will be forgotten when discussing the results, while this potentially weakens the confidence in the results from both methods; but this inter-annual change at the national scale may be negligible compared to the seasonal, day-to-day and diurnal variations biasing the comparison between annual budgets in NAEI and the flux estimates for daytime in March (see my comment above)

* section 3.1.1

-p6l22-23: The sentence (especially "enabling us to specifically assess accuracy...") seems to ignore the significant fluxes upwind and downwind London; see my point (2)

-p6l31: the latitudinal gradient is not fully accounted for since the background to be removed from local concentrations is taken as a constant value (the average between the north and south backgrounds) rather than as a linear interpolation between the north and south backgrounds; these north and south backgrounds sometimes seem to strongly differ: isn't it an issue (at least as significant as the one raised on p7l1-2) ?

* section 3.1.2

- p7l17: do the measurements and/or simulations show a significant change of vertical gradients in the concentrations when crossing this BLH ~750m (it does not seem to be the case in Figure 2) ? does the vertical distribution of the concentrations say something about the reliability of the model ?

- p7l29: see my main point (2), you need strong assumptions to apply the scaling factors derived for a large part of the South of England to the Greater London area.

- p7l33: I think that this statement is a bit extreme, especially since several investigations could be led to provide insights on the transport uncertainties: the analysis of the wind fields (see my main point (3)), of the 2D vertical structure of the concentration measurements, and, maybe, of the measurements around the Greater London area that are not exploited in this study

- p8l20-23 are a bit confusing. I do not really catch how the spatial distribution will be tackled along with the temporal variability.

- p8l30: the human respiration could also be listed as a source of mismatch ?

- p8 in a general way: the authors should try to better connect and discuss together the results for CO and CO2: why the scaling factors are so different for these two species ? is it due to the natural CO2 fluxes only ? would not it say something about these natural fluxes ?

- p8l31-32: "we can expect them to underestimate" -> shortcut

* section 3.2.1

-p9l30: "horizontal boundaries" could be rephrased for clarity. Could the definition of the background as the average concentrations over the 15-km boundary sections be too crude for focusing the emission estimate on the Greater London area (is the 15 km distance too short) ? does this background fit well with the background estimated with the flux dispersion method ?

-p10l1-3: the discussion goes a bit too fast for me. One could also assume that the up-wind concentrations are more suitable to define a background for the measurements downwind London because they would characterize a section across their footprint that is relatively close to the sea (Figs 1 and 4). Discussing the impact of BLH on background concentrations could mean that these background concentrations are mainly driven by fluxes that are relatively close to the measurements. However, the concentrations North and South of the transects A-B are mostly influenced by fluxes North and South of London that are hardly seen by the measurements downwind London, as indicated by Figure 4. In a more general way, I think that the characterization of the "background concentrations" and footprint for the measurements downwind London could be better discussed (see my main point (2)).

References:

Brioude, J., Angevine, W. M., Ahmadov, R., Kim, S.-W., Evan, S., McKeen, S. A., Hsie, E.-Y., Frost, G. J., Neuman, J. A., Pollack, I. B., Peischl, J., Ryerson, T. B.,

Holloway, J., Brown, S. S., Nowak, J. B., Roberts, J. M., Wofsy, S. C., Santoni, G. W., Oda, T., and Trainer, M.: Top-down estimate of surface flux in the Los Angeles Basin using a mesoscale inverse modeling technique: assessing anthropogenic emissions of CO, NOx and CO2 and their impacts, Atmos. Chem. Phys., 13, 3661-3677, https://doi.org/10.5194/acp-13-3661-2013, 2013.

Font, A., Grimmond, C. S. B., Kotthaus, S., Morguí, J. Â■A., Stockdale, C., O'Connor, E., Priestman, M. and Barratt, B. (2015) Daytime CO2 urban surface fluxes from airborne measurements, eddyÂ■ covariance observations and emissions inventory in Greater London. Environmental Pollution, 196. pp. 98Â■106. ISSN 0269Â■7491 doi: https://doi.org/10.1016/j.envpol.2014.10.001.
* * *

---

## Author Comment (AC1) · 22 Feb 2019

**Author response to referee comments**

We would like to thank both referees for their constructive comments and insight. They have provided us with plenty of food for thought, both in terms of the tone of the paper and its place within the wider context of the field. Below we respond to each referee individually, however as both referees raised similar points in many places we have combined our replies into a single document.

**Referee 1:**

Pg 1 ln 19-20. Consider rephrasing – many in the atmospheric greenhouse gas community are recognizing that the value of atmospheric measurements in the emissions reporting context is in working with existing inventories to evaluate and improve emissions reporting. Presenting these measurements as "independent verification" pitted against inventory methods is problematic.

A good point – amended to better emphasise the potential synergy between top-down emission estimates and bottom-up inventories

Pg 2 ln 3-10. Please add some discussion here about sources/sinks that are not included in the NAEI. For CO2 this is mainly biogenic fluxes, which are noted later to be critically important. Please also include a discussion of what sources of CO and CH4 are included in the inventory, and which are not. For example, I suspect that oxidation of biogenic VOCs is not included in the CO inventory. These may be negligible in March, but should still be mentioned. As for the CH4 inventory, does it include all sources, or only anthropogenic sources, and how significant might non-anthropogenic sources be?

A brief paragraph on the natural sources of these gases has been added. For $CH_4$ and CO the impact of natural sources on the results of this study is likely to be small, but as discussed later in the paper this is not the case for $CO_2$. Rather than repeat it here, we refer the reader to the relevant section for this discussion.

Pg 2 lines 21-22. While it is true that comparison of top-down estimates with bottom up inventories is one important way to use the atmospheric observations, it is certainly not true that the only use of these measurements is to evaluate inventories! Please rephrase.

We certainly didn't intend to imply that the measurements themselves were only useful for this purpose, only that the mass balance flux calculated through the downwind sampling plane is not particularly informative in isolation. This value, which physically represents the mol s$^{-1}$ passing through some arbitrary vertical plane above some defined background mole fraction, needs to be related to the emissions for a given city/region/source for it to have any meaning. The lack of clarity in this section was also highlighted by referee 2 – it has been edited to address both comments.

Pg 4 ln 25. "an altitude-latitude plane."

Corrected

Pg 5 ln 6-20. Please add a sentence that explains in plain English the principle of what the equations do, rather than requiring the reader to wade through the equations to figure out the principle (although the detail of the equations is necessary too).

We've reordered this section to try and make it flow better.

Pg 5 section 3. Please add some detail about the NAEI. It is spatially explicit, but does it have temporal variability? If so, what kind of temporal variability and how reliable might that be? Diurnal cycles? Seasonal cycles? Weekday/weekend? Are there any existing estimates of the quality of the

inventory (and perhaps the quality is different for the different gases)? This becomes important in trying to understand the differences between the inventory and the observations.

The NAEI only contains annual averages – we've added a sentence to explicitly state this and alluded to the fact that this is likely to be a source of model-measurement disagreement for a single-flight case study.

Pg 6 ln 24-25. Again, please add a sentence that explains the principle in plain English rather than forcing the reader to work it out from the equations. Eg "The mole fraction enhancement is calculated by subtracting the background value".

Added

Pg 6 ln 24-32. The choice of background is known to be a key uncertainty in this type of measurement (eg. Cambaliza et al 2013; Heimburger et al, 2017). Unfortunately the research community has not yet come to any conclusion as to how to resolve this. The simple method of taking an average of the values measured on the downwind edges of the plume (as is done here) is far from perfect, even if it might be the best available option given the measurements that have been done. Heimberger et al (2017) showed that there can be significant differences in the values on the two edges, and that in that case, a simple improvement would be to linearly interpolate between the two edges to evaluate background. It is also entirely possible that the background is not uniform and that there are plumes from upwind sources that are not detected because they are inside the urban plume. From Figure 4, it's apparent that there are a lot of methane emissions upwind of the city that could cause this. Further, there's an implicit assumption that there are no emissions occurring in the footprint of the edge measurements. This is clearly a bad assumption for this dataset, and so the edge measurements will be biased high (or perhaps low in the case of CO2 if there is significant drawdown in the edges), resulting in an underestimate of the urban emission rate (or perhaps overestimate in the case of CO2). A forthcoming paper (in last phases of review) will discuss this further, but unfortunately is unlikely to be published in time to be referenced in this paper. My suggestion is to: (1) Add a figure that shows clearly the background values, how they were chosen, and whether there is any difference between the two edges. (2) A plot of the upwind measurements could also be included to show whether there is any particular concern with plumes coming in from upwind for this dataset. (3) Add figures that show the NAEI CO2 and CO emissions, similar to that shown in Figure 4 for CH4, to give a sense of upwind and edge emissions and how important they might be. (4) If there are no particular concerns with the points above, then stick with the current choice of background. (5) Add some discussion about the uncertainty associated with choice of background and how it might influence the results.

We absolutely agree with the referee that this is the key issue with the mass balance method – this is what motivated us to develop the flux-dispersion method which we think is less susceptible to the biases it can cause. The fundamental problem with applying the mass balance method here is that there are many emission sources outside London that contribute to mole fraction enhancements in the downwind plane. Hypothetically, to deal with this issue in the context of the mass balance method, one either needs to account for the influence of these emissions in the background mole fraction (such that all downwind enhancements are a result of Greater London emissions) or include them in the aggregated inventory emission total against which the top-down flux is compared. Our original draft focussed on the difficulties associated with the latter approach – we have redrafted Sects. 3.2 and 3.3 to include discussion on the former approach too.

One thing that is worth clarifying regarding the comment made here is that Pg 6 L24-32 describes the approach taken in the flux-dispersion method. However the issues raised here by the referee are more relevant to the mass balance method discussed in Sect. 3.2. The flux-dispersion method does take into account emissions in the footprint of the edge measurements, and it is made explicit in this section that the results obtained using this approach pertain to the emissions from the areas sampled in Fig. 4b relative to the emissions sampled in Fig. 4a.

We have repeated the flux-dispersion analysis with an interpolated (rather than averaged) background value used across each transect, and each ratio changed by less than 0.01. We have made reference to this in Sect. 3.1.1. Similarly we repeated the mass balance method with interpolated background values and found this slightly increased the derived fluxes, but in all cases the difference was less than 7%. We have included these values in Sect. 3.2.2.

On the specific points made above:

1) We have added a table containing the background values used in the flux-dispersion method as we thought this shows them more clearly than a figure.

2) Unfortunately the upwind sampling was nearly all either out of the boundary layer or dipping in and out of the top of it, so this data is not really useful for defining a background here.

3) These plots have been added to Fig. 4.

4) As the flux-dispersion method accounts for the emissions in the background footprint, the key to defining the background is to set a criterion that ensures emission from the area of interest (in this case Greater London) represent the most significant difference between the in-plume and background footprints. Comparing Fig. 4a and Fig. 4b we feel that the background choice here achieves this. This is discussed further in our response to Referee 2 below.

5) We have clarified the discussion in Sect. 3.1.1 of how the choice of background influences the flux-dispersion results, emphasising the impact of this choice on the selectivity of the results towards London. For the mass balance method we have added discussion on the choice of background to Sects. 3.2 and 3.3.

Pg 7 ln 24-30. Looking at figure 5, there's a clear spatial mismatch in the plume location between the obs and simulation. What might be the explanation for this? Given this mismatch, is it reasonable to average over the whole thing and then compare the two methods? This mismatch seems to imply a larger uncertainty than that given by just comparing the means.

This is a good point that deserves discussion. There are two obvious potential causes:

1) The spatial distribution of the inventory is incorrect, such that for all three species it underestimates emissions from south London and/or overestimates emissions from north London.

2) The spatial distribution of emissions in the inventory is broadly correct, but inaccuracy in the model wind field causes the simulated plume to be advected to a slightly more northerly position.

Cause 2) seems to be the more plausible, given that the simulated plume is north of the measured plume for all three species, despite the different source mixes for each species (in particular $CH_4$). On the other hand, the model wind direction compares fairly well to the measured winds at the aircraft sample locations, and any disagreement between them is actually in the opposite direction (i.e. the

measured winds are more southerly). However, it still seems like too much of a coincidence for all three species to be incorrectly distributed in such a way to generate exactly the same offset in plume position, so we believe inaccuracies in the UKV wind field (prior to reaching the sampling plane) are indeed the likely cause of this mismatch.

The uncertainty associated with the model wind field is one of the main points raised by Referee 2 – please see below for a discussion of this.

Pg 8. Please emphasize throughout the discussion of the comparison that this analysis is for a single flight, and that care should be taken in drawing conclusions about the integrity of the inventory from a single comparison on a single day. Previous authors have shown that when multiple flights are considered, there can be large differences in the calculated flux that are likely due to uncertainties in the top-down flux estimate rather than day-to-day differences in the actual emissions.

Section edited to emphasise this

Pg 8 ln 10-23. I agree that an incorrect spatial pattern in the inventory could explain at least part of the difference. However, I suspect that the choice of background may be more important and be biasing the top-down estimate low. See earlier comments. Does the NAEI include temporal variability and could lack of temporal variability in the NAEI be an explanation for the difference? See earlier comment.

We have edited this section to emphasise the likelihood that temporal variability in emissions (which the NAEI doesn't include) is a likely candidate for this difference. See above for our discussion regarding the choice of background – this can clearly bias the mass balance emission estimates but the flux-dispersion method takes sources within the background footprint into account. Biases associated with inaccuracy in simulated background footprint are discussed at the beginning of Sect. 3.1.2.

Pg 8 ln 24 – 35. It's clear than biogenic $CO_2$ will have an enormous influence on the calculated flux, and that it can bias the $CO_2$ background quite dramatically (see e.g. Turnbull et al 2015, Cambaliza et al 2013). The statement here needs to be much stronger, "treated with caution" is an understatement! It is simply not possible to compare a flux based on total $CO_2$ with an anthropogenic $CO_2$ inventory unless the biogenic component can be accounted for, likely by either having a good biogenic model or being able to separate biogenic and fossil fuel $CO_2$ in the observations (e.g. using 14C or CO). I would say something like "comparison with the NAEI is not appropriate for this dataset".

Edited to make statement stronger

Pg 9 ln 28-34. See earlier comments about choice of background. The same biases occur for this method as for the other method.

This point is discussed in response to the earlier comment

Pg 10 Section 3.2.2. Can you come up with a total emission flux for the flux dispersion method, so that the total flux from each method can be compared more directly? As written, the comparison is between the obs/model ratio for each of the two methods. Thus it can't be determined whether the difference in the ratio occurs because the observed flux rate is different, or the modeled flux rate is different. You argue that the difference is in the modeled flux (actually that you've defined the modelled footprint differently in the two cases). By making a slightly different comparison, this could be argued more strongly.

When we initially started developing the flux-dispersion method, our first approach was to krig the simulated mole fractions and compare total kriged fluxes through the downwind plane. However, we moved away from doing this because by definition any difference in the flux ratios derived in this manner from the overall flux ratios calculated here purely results from the interpolation/extrapolation of sparse data. While the overall flux ratios calculated here are essentially the average of individual transect flux ratios weighted by flux density enhancement, using the kriged flux ratios this average is weighted by a somewhat arbitrary factor relating to the location of the transects relative to each other on the sample plane. In any case, returning to this approach would not tackle the question posed above, because it uses the measured flux calculated with the mass balance method (i.e. the modelled flux is the only value that changes between the two methods).

We have made changes throughout Sect. 3.2 and 3.3 regarding the relationship between the choice of background and the choice of inventory aggregation area (i.e. the "modelled flux") for the mass balance method. Hopefully this explains better the issue with the application of the mass balance method in this study.

Pg 10 section 3.3. This difference in how the footprint is defined is a good candidate for the difference. There are potentially ways to resolve this in the mass balance method. A good start would be to make an estimate of the footprint of the mass balance, rather than assuming that the footprint is an arbitrary metropolitan boundary.

It is not clear to us how an unambiguous definition could be made which separates emissions that contribute to the plume from emissions that contribute to the background (as many source areas contribute to some extent to both). This is discussed at length in response to Referee 2's main point 2).

Pg 11. Conclusions. Please restate the point that the CO2 comparison is invalid because biogenic CO2 is not accounted for. Otherwise the conclusions are very nice.

Point added

Figure 1. Please add a larger scale inset to show where this is in relation to the UK, Ireland, etc. Not all of us are well-versed in English geography!

Inset added

**Referee 2:**

1) My understanding is that the new approach is merely a combination or trade-off between two traditional approaches : the mass balance approach and the atmospheric transport inversion (Brioude et al. 2013 provide an example of inversion applied to aircraft data around a city). Conceptually, the major difference between this approach and the traditional atmospheric transport inversion is related to the fact that the observed variables to be fitted by rescaling the surface fluxes are fluxes at the measurement locations rather than concentrations. This requires some additional assumptions for the computation of such fluxes, but this enables to account for wind measurements when assimilating the observations. Another difference is that rather than assimilating all local fluxes at the aircraft measurement locations in a Bayesian statistical inversion framework, the method consists here in summarizing them into an average value which is used to rescale the map of surface fluxes. This simplification could lead to a loss of information but it can also help control the inversion behavior. One of the strength of traditional atmospheric inversions and of this new approach is the ability to

extrapolate the information from the sparse measurements by accounting for the atmospheric transport and for the emissions spatial distribution, while the traditional mass balance approach makes coarser extrapolations (here based on a kriging technique).

I think that such a comparison to the atmospheric inversion is worth being discussed since the comparison to the mass balance approach only could lack of hindsight regarding the panel of methods that have been tested to exploit aircraft data. Furthermore, from my point of view, this new approach is closer to the atmospheric transport inversion than to the mass balance approach.

This is a very fair point – in many ways this technique is more similar to previous transport inversion studies than it is to those that use a mass balance method. The focus on mass balancing in the introduction probably reflects the process we went through in developing the technique: we started with a conventional mass balance calculation, but in attempting to resolve the issue of surrounding emission sources each iteration of the technique increasingly revolved around use of the NAME air histories.

We have edited the introduction to better place this method in the context of other approaches used, particularly with regard to atmospheric transport inversions. Flux estimation using a full atmospheric transport inversion is beyond the scope of this study, however a separate study which includes emission estimates for the GAUGE flights derived using a trace gas inversion (employing a hierarchical Bayesian framework) is currently in preparation.

2) One of my main concerns is that by rescaling the total of the NAEI emissions according to measurements whose surface footprint extends well beyond the Greater London area, the new approach does not really inform on the emissions from this area either. Given the distances from the section A-B to London, and as illustrated by Figure 4, the results from this method are driven by emissions from a large part of the South of England that extends to the sea, despite the removal of the "background" concentrations (whose sensitivity to the Western part of the South of England seems much smaller than that of the measurements used to constrain the estimate of emissions according to Figure 4).

Yes, a disadvantage of this new technique is that it is not entirely selective of emissions from a single source area. The key to achieving results relevant to the area of interest is to choose appropriate criteria to define the background, as described in Section 3.1.1. Using the contours in Fig. 4c as a guide it appears reasonable to claim that emissions from the London conurbation represent the most significant influence on the difference between the in-plume and background fluxes, thus supporting the choice of background threshold used here.

This point highlights the key issue with applying the mass balance method in this case, as similarly this is not selective of Greater London emissions but is influenced by emissions from a much wider (but ill-defined) area. In this case the lack of selectivity actively biases the results, an issue which is resolved by applying the new flux-dispersion method. However that both methods suffer from this lack of selectivity was not obvious enough in the original paper – this has now been edited throughout to make it clear.

The computations are conducted in March so that ignoring the natural $CO_2$ fluxes might be fine. But similar computations based on the same aircraft campaigns in spring and summer would be highly hampered by the $CO_2$ uptake upwind and downwind London (not only by the differences between the natural fluxes within the urban part of the measurement footprint vs. within the background footprint that are discussed in section 2.3). While the lack of account for natural $CO_2$ fluxes is mentioned in

section 3.1.2, the major issues raised by these fluxes for spring / summer deserve a discussion, and the topic could deserve some indications in the method sections (in particular in section 2.3) and maybe a coarse look at estimates of the CO2 natural fluxes in the UK.

Yes natural fluxes would be much more significant in summer, and probably cannot be taken as negligible here. We have edited the paper to make a stronger statement regarding the inappropriateness of direct comparison with the NAEI for $CO_2$ (this point has been particularly highlighted by Referee 1) and have added a paragraph discussing the inferences that can and cannot be drawn regarding natural fluxes by considering the results for $CO_2$ and CO together (more on this in response to the specific point raised below).

I feel that the manuscript is a bit severe with the mass balance approach by crudely attributing the flux estimate from this method to the Greater London area, and maybe by deriving an estimate of the background concentrations for this approach in a crude way. More cautious interpretations of the flux estimates from this approach are usually made, especially for situations like that of London. I would recommend the authors to comment on the paper by Font et al. (2015) who also made estimates of the emissions from London using aircraft data, and who used FLEXPART simulations to assess the footprint of their measurements. O'Shea et al. (2014) also used NAME to analyze the footprint of their aircraft measurements, and discussed the issue that would be raised by the crude assumption that these measurements would correspond exactly to the greater London area.

It is certainly true to say that aggregating the emissions over the Greater London area is a crude approach. However, it is not clear to us that a more robust method for defining this footprint exists. Font et al. (2015) are able to define a footprint for their measurements using FLEXPART, but their Integrative Mass Boundary Layer method relies on a completely different calculation to the mass balance method used here. They essentially measure the difference between the rate of change in $CO_2$ concentration within the boundary layer and the rate of entrainment of air from above. This yields a surface flux per unit area, which can then be related to the area the air has travelled over using FLEXPART. In the mass balance approach used here we calculate a bulk flux (i.e. not per unit area) through the downwind sampling plane, relative to our choice of background mole fraction.

The problem here is that, while emissions from certain areas clearly contribute only to the plume, and emissions from certain others contribute only to the background, the majority of the emissions over the air history contribute to both but to different extents. This can clearly be seen in Fig. 4 – summing all of the emissions covered by the air history in Fig. 4b would clearly overestimate the calculated flux (even if they are weighted by residence time, as in Font et al., 2015), as most of these areas are also covered by the air history in Fig. 4a and so contributed to the background. Although this figure relates to the in-plume/background periods for the flux-dispersion method (which is slightly different to the background used in the mass balance calculation taken from the kriged plane), it demonstrates the difficulty in defining any objective criteria for determining the aggregation area. One could subtract the background air history from the in-plume air history and define some threshold value above which a grid square would be included in the aggregated flux total, but the choice of such a threshold would be just as arbitrary as simply using the Greater London boundary. We have added a discussion of this to Section 3.3.

O'Shea et al (2014) took a different approach, using NAME to discard cells from the kriged plane which didn't contain measurements of air that had passed over Greater London. In other words, they stick to aggregating bottom-up emissions over the Greater London area, but try to downscale the kriged flux to represent this area. However, this doesn't solve the fundamental problem with the mass

balance approach, as grid squares with an influence from Greater London can still be influenced by sources surrounding Greater London as well. In the case study presented here, the in-plume measurements were all influenced by London to some degree (as can be seen by comparing Fig. 2 and Fig. 3), so this downscaling approach is not easily applicable.

Therefore, I would be ready to agree that the mass balance approach and its associated type of aircraft measurement tracks is not very well adapted to the monitoring of the emissions from a city surrounded by other cities and productive ecosystems, especially if flight regulations impose measurements to be conducted far downwind. However, I feel that by relying on the same type of measurements and by avoiding to solve for the spatial distribution of the emissions, the new approach may bear the same fundamental limitation which is the lack of ability for isolating the budget of the emissions from the targeted city. In this regard, I think that the conclusions are a bit optimistic.

An additional caveat has been added to the conclusion section on this point.

3) A critical variable in the study is the wind which is used to compute fluxes. Comparisons between measured and modeled (UK Met Office) winds along the transects but also all around the London Greater area could potentially provide some strong insights on the robustness of the transport model, of the estimate of the measurements spatiotemporal footprint and of the estimate of the surface emissions (in particular if biases arise in the comparisons). I feel that it deserves some analysis.

The wind field does indeed have a strong influence on the flux, which is why we have adopted the approach here of taking flux density ratios (rather than concentration ratios). Within the boundary layer the modelled wind speeds are generally higher than the corresponding measurements, both upwind and downwind of Greater London. We account for this by using the modelled wind speed to calculate the simulated flux density, on the assumption that the model overestimation of wind speed will result in a corresponding underestimation in simulated concentration enhancement. We have expanded the discussion of this point in Sect. 3.1.1 and included a figure showing both model and measured wind speeds throughout the flight.

The impact of the overestimated model wind speed on the spatial extent of the footprint is far more subtle and difficult to account for. Given the high bias of the model wind speeds, it is reasonable to assume that the air history likely underestimates the spatial spread of the sample footprint, resulting in the in-plume measurements having a higher simulated sensitivity to emissions from Greater London. This in turn could introduce a low bias into the inventory scale factors.

Any inaccuracy in wind direction across the model wind field also clearly contributes to the uncertainty in this method. Referee 1 points out the mismatch between simulated and measured plume position, which is likely to be a consequence of inaccuracy in the model wind field. This too highlights a potential source of bias, as the air histories for in-plume and background periods simulated by NAME may differ slightly from the actual air histories of the measurements.

The obvious way to investigate the impact of wind field inaccuracy on the uncertainty budget would be to conduct a sensitivity test, where an ensemble of NAME runs are performed driven by a set of met data with perturbed wind fields. This would be an interesting study, but is quite involved from a modelling perspective and goes well beyond the simple use of NAME in this study. We have added a discussion of this potentially significant source of uncertainty in Sect. 3.1.2.

**Detailed comments:**

- p2l1: explain that "top-down" relates to methods based on atmospheric measurements and models ?

Explicit statement of this added

- p2l5-6: do power plant represent a large fraction of the CO2 emissions in the greater London area ? on the same topic: I had in mind that the city had large power plants in its vicinity that could represent a major share of the emissions in the measurement footprint (http://naei.beis.gov.uk/data/gis-mapping): is it the case ? if yes, it would feed my main concern (2)

Within Great London power plants represent a small fraction of the emissions (<5%). However, power plants in the areas surrounding Greater London do constitute a significant source of emissions, with power plant emissions (including energy from waste facilities) summing to over 25% of the Greater London total. So the referee's main point 2) is valid (see discussion of this above) – it is not possible to isolate the Greater London from surrounding sources using either method presented here, and we have altered the phrasing of the method discussion to clarify this. However, this point also emphasises the advantage of the flux-dispersion method over the mass balance method, as fluxes outside Greater London actively bias the mass balance method results, whereas these surrounding power plant emissions are accounted for in the flux-dispersion method (their presence only acts to reduce the selectivity of the results to the Greater London region).

- p2l15: I am not sure about the meaning of "bulk area flux" here. What would prevent atmospheric inversion to provide such a bulk area flux based on the same data ? see my main point (1)

Yes looking back we agree this is a false distinction – this sentence has been removed anyway in response to main point 1.

- p2l21-33: I feel that the problem of defining the footprint of the estimated flux is presented in an "inverted" way which makes things more complicated than they are. In particular there is no reason to necessarily involve inventories in this problem.

This paragraph has been edited to address the comments of both reviewers. We agree that the emphasis was previously placed too strongly on the comparison with inventory fluxes – the fundamental issue is that, for non-isolated cities, the top-down flux cannot easily be ascribed to a well-defined region. This in turn does make it difficult to compare bottom-up and top-down fluxes using the mass balance method – we have tried to make this narrative clearer in the revised text.

* section 2.2 - this section should provide the duration and the period of the day corresponding to the flight. Maybe I missed it in the following, but the time of the measurements is a critical information that can raise questions regarding the temporal representativity and the robustness of the computations

Requested details added – we agree that this helps to interpret the flux values derived in subsequent sections in the context of the diurnal variability of these fluxes.

- p4l11: I do not understand the end of the sentence ("so as to assess the representativeness ...") in its context

Changed "representativeness" to "accuracy" to clarify what is meant here.

- p4l13: "less than 24 hours" -> 24 hours is large if considering the need to connect the measurements to an emission footprint both in space and in time, and given the strong diurnal variations of the fluxes. Can the statement be more precise based on NAME simulations ?

This has now been made explicit as an average transit time of 20 hours over the British Isles. It is true that diurnal variability in fluxes over this timescale could effectively weight the sensitivity of the derived flux ratios to regions the air passed over during times of peak emission (e.g. rush hour). However, from Fig. 4 it can be seen that the aggregate air histories for in-plume and background sampling begin to converge upwind of London, and certainly by the time the particles leave the domain there is little discernable difference between the distribution of particles for each period. Fluxes in regions that contributed equally to both background and in-plume measurements have no effect on the results, so despite the fact the average transit time over the British Isles is 20 hours, the results are sensitive to fluxes over the much shorter transit time before the two aggregate air histories converge.

* section 3 (beginning) -p5l30: maybe you should already clarify here the fact that NAEI provides annual budgets of the emissions only, while the measurement were made during day time in March, which corresponds to a period of relatively high emissions (this information is limited to the discussions on the CH4 results, and just ignored for CO and CO2 in section 3.1.2). Using constant emissions in the model may also be problematic because the duration of the measurement campaign is about 2.5h, during a period of the day when emissions could be highly variable.

Explicit statement of this added

-p6l1-3: this will be forgotten when discussing the results, while this potentially weakens the confidence in the results from both methods; but this inter-annual change at the national scale may be negligible compared to the seasonal, day-to-day and diurnal variations biasing the comparison between annual budgets in NAEI and the flux estimates for daytime in March (see my comment above)

Point added

* section 3.1.1 -p6l22-23: The sentence (especially "enabling us to specifically assess accuracy. . .") seems to ignore the significant fluxes upwind and downwind London; see my point (2)

Agreed this statement was too strong. This section has been amended in line with point 2) to clarify that, while one can choose a threshold to optimise the selectivity of the results towards the region of interest, there will inevitably be some influence from a wider region.

-p6l31: the latitudinal gradient is not fully accounted for since the background to be removed from local concentrations is taken as a constant value (the average between the north and south backgrounds) rather than as a linear interpolation between the north and south backgrounds; these north and south backgrounds sometimes seem to strongly differ: isn't it an issue (at least as significant as the one raised on p7l1-2) ?

We have repeated the flux-dispersion analysis with an interpolated (rather than averaged) background value used across each transect, and each ratio changed by less than 0.01. We have made reference to this in the paper. See also the discussion of this section in our reply to Referee 1 above.

* section 3.1.2 - p7l17: do the measurements and/or simulations show a significant change of vertical gradients in the concentrations when crossing this BLH ~750m (it does not seem to be the case in Figure 2) ? does the vertical distribution of the concentrations say something about the reliability of the model ?

The vertical distribution of concentrations outside the plume looks similar in the simulated and measured datasets. However, the simulated flux density enhancements are smaller relative to the measured enhancements when sampling above the model boundary layer height. As the simulated dispersion above the model boundary layer height is known to be less accurate we did not include this data.

A plausible explanation for underestimation of the simulated plume magnitude above 750 m would be if the simplified turbulence parameterisation above this height led to the suppression of vertical mixing in the model. In such a case the simulated plume magnitude within the model boundary layer could be consequently overestimated. A full investigation into the impact of turbulence parameterisations on the vertical mixing within the NAME model would require a separate study, but we have added a brief discussion raising this issue as a potential source of bias.

- p7l29: see my main point (2), you need strong assumptions to apply the scaling factors derived for a large part of the South of England to the Greater London area.

Sentence added to re-emphasise the influence of surrounding sources on the results.

- p7l33: I think that this statement is a bit extreme, especially since several investigations could be led to provide insights on the transport uncertainties: the analysis of the wind fields (see my main point (3)), of the 2D vertical structure of the concentration measurements, and, maybe, of the measurements around the Greater London area that are not exploited in this study

Yes fair point – an ensemble NAME run using perturbed met data could be used to quantify some of the uncertainty associated with the dispersion modelling. As stated above, we feel this is beyond the scope of the study here, but we have expanded the discussion of dispersion model uncertainty and highlighted this as a topic for future study.

- p8l20-23 are a bit confusing. I do not really catch how the spatial distribution will be tackled along with the temporal variability.

This section has been reworded to make it clearer. Essentially we are trying to stress the point that more flights would be required to capture temporal emission variability (and reduce the impact of random errors).

- p8l30: the human respiration could also be listed as a source of mismatch ?

We have added explicit reference to this as a component of the higher heterotrophic respiration within the city.

- p8 in a general way: the authors should try to better connect and discuss together the results for CO and CO2: why the scaling factors are so different for these two species ? is it due to the natural CO2 fluxes only ? would not it say something about these natural fluxes ?

We've added a paragraph discussing this. The differences in emission ratios for several key sources makes it a bit of a stretch to attribute all of the difference to the net biospheric flux, but it provides a useful ballpark guide to the potential magnitude of these fluxes.

- p8l31-32: "we can expect them to underestimate" -> shortcut

Changed to "we expect them to underestimate"

* section 3.2.1 -p9l30: "horizontal boundaries" could be rephrased for clarity. Could the definition of the background as the average concentrations over the 15-km boundary sections be too crude for focusing the emission estimate on the Greater London area (is the 15 km distance too short) ? does this background fit well with the background estimated with the flux dispersion method ?

"Horizontal boundaries" has been rephrased. The definition of the background here is crude, but as per the footprint it isn't clear that a more robust method to define where the plume ends and the background begins exists. Simply eyeballing the data and selecting where the plume ends is a fairly typical approach used in many previous studies. Other studies use some statistical threshold (e.g. 3σ less than the mean mole fraction), but these are in essence equally arbitrary. The average background values used are quite close to those used in the flux-dispersion method – we've added these values to the text.

The alternative approach of interpolating (rather than averaging) the background values has been tested. This is discussed in our response to Referee 1 above. We have expanded the discussion on uncertainties associated with the choice of background in Sects. 3.2 and 3.3. See also the response to the final comment below.

-p10l1-3: the discussion goes a bit too fast for me. One could also assume that the upwind concentrations are more suitable to define a background for the measurements downwind London because they would characterize a section across their footprint that is relatively close to the sea (Figs 1 and 4). Discussing the impact of BLH on background concentrations could mean that these background concentrations are mainly driven by fluxes that are relatively close to the measurements. However, the concentrations North and South of the transects A-B are mostly influenced by fluxes North and South of London that are hardly seen by the measurements downwind London, as indicated by Figure 4. In a more general way, I think that the characterization of the "background concentrations" and footprint for the measurements downwind London could be better discussed (see my main point (2)).

We have edited and expanded this section to better discuss the pros and cons of each method for determining the background. Ideally we would define the background so that it represents the mole fractions throughout the downwind plane that would be measured in the absence of any sources from within Greater London. Were we able to calculate this hypothetical value it would solve the issue regarding the definition of aggregation area – if all extraneous sources were accounted for in the background then all enhancements in the downwind plane would only emanate from emissions within Greater London, so aggregating inventory emissions within the area would be reasonable. So, as discussed in our response to Referee 1, the issues raised here with the background definition is essentially another way to express the main issue regarding the inventory aggregation area raised in this paper. We have edited the discussion in Sects. 3.2 and 3.3 to cover this.

---

## Author Response (AR2)

**Author Response**

We thank the reviewer again for their thoroughness – on returning to the manuscript we agree that it lost some of its flow during the redraft. In particular, the discussion of extraneous emission sources outside Greater London was poorly focussed and spread across multiple sections. We have made further edits following proofreading, including responses to the specific points raised below.

With the major revision of the discussions, the manuscript « Assessing London CO2, CH4 and CO emissions using aircraft measurements and dispersion modelling » by Pitt et al. is now much more cautious and balanced when assessing the results from their estimate of the CO2, CH4 and CO emissions from the Greater London area.

I am fully satisfied with the answers to my comments on its first version. The transcription of these answers in the manuscript is not always as convincing: in particular, I think that the abstract could be more significantly updated in line with the new major discussions. However, all the issues are now discussed somewhere and the authors made a clear point about the lack of confidence on the CO2 estimate.

Slight edit to the abstract in line with the fact that the results are not only sensitive to emissions within Greater London itself.

My concern is now that the additions to the text often seem to have been written quite quickly. As a result, the paper has lost a bit of its quality and its structure has been a bit weakened. I recommend a new in depth proofreading to the authors and maybe a bit of re-ordering. Here is a list of minor comments, most of which relate to this:

- Some of the references to other sections are a bit cycling and there are now redundancies, because the discussions have been split and spread among several sections rather than addressed in depth at a relevant location. In particular:

-> regarding the influence of fluxes upwind and downwind the Greater London area: p13 l21 in Sect. 3.2.1 refers to Sect. 3.3 and p14 l9 in Sect. 3.3 refers to Sect 3.2.1 and both actually discuss the topic in the same way; this topic is also discussed in Sect 3.1.1 in which p8 l2 refers to Sect. 3.3. and in Sect. 3.1.2 (« It is worth stating again ») in which p10l18 refers to 3.1.1. Finally, the end of the introduction (p3 l26-28) surprisingly gives a first summary of the results and already starts to discuss this topic.

We agree that this was a major weakness of the revised manuscript. A full discussion of this issue is now presented in Sect. 3.3, incorporating the various elements previously scattered around the paper (in particular those points previously discussed in Sect. 3.2.1). We have removed the repeated statement from Sect. 3.1.2 ("It is worth stating again...").  The point regarding the introduction is also pertinent – we have removed the summary of results from the end of the introduction, and also reduced the paragraph p2 l26 – p3 l2 (which effectively discussed the conclusions of the paper) into a short summary outlining the issue.

-> the strategy for measuring the background is discussed in details in Sect. 3.2.1 but not in Sect. 3.1.1 which just states « given the sampling strategy employed » on p7 l29-30

We have added a reference on p7 to the upcoming discussing in Sect 3.2.1. While in some ways it would be preferential for this discussion to come in the first of these two sections, we feel it sits much better within the discussion of the mass balance method (among other reasons because all of the studies referenced in this discussion are mass balance studies).

-> the topic of the influence of natural fluxes has been disconnected from that of the upwind and downwind fluxes (see Sect. 1 and Sect. 3.1.2) but they are strongly connected

To some extent these are connected, in that many natural flux sources are outside the Greater London boundary. However we believe that they are fundamentally different issues; even if there were no fluxes in the areas surrounding Greater London, natural fluxes within the Greater London region (which for $CO_2$ are significant) would still impact on the interpretation of the results. The influence of natural fluxes is fundamentally a limitation of the inventory we are using, so it is included in the discussion of the NAEI. The influence of sources outside the Greater London region is a methodological issue, so we are not convinced that these discussions would be better merged.

-> regarding the temporal representativity of the estimates: p7 l4 in Sect. 3 refers to Sect. 3.1.2 but p7 l1-4 already give the main details of the discussion in Sect. 3.1.2, and Sect. 3.1.2 even says on p10 l27 « it is important to remember ». p11l3 seems to start the discussion once again, ignoring the beginning of the previous paragraph (these two parts should be merged).

Reviewer 1 requested that the timescales represented in the NAEI were explicitly stated in Sect 3 (p7) – we think it makes sense to bring this up at the beginning of this section so people are aware of it before they get to the results. The repetition between p10 and p11 arises from discussing the issue in the context of the CO results then revisiting it when we get to the $CH_4$ results. We agree this is a bit laborious and have reworked this section to avoid this repetition.

-> discussions on p8 l17-28 and on p9l29 to p10l9 may work better merged together

Agreed – sections merged

- the comparison to the atmospheric inversion should be slightly revised:

-> p3 l10: « over constraining such that the results only reflect the choice of prior emissions »: if the atmospheric observations over constrain the problem, the weight of the prior should be low, did the authors mean « under constraining » ?

-> p3 l11-12: I am not sure to understand what it means, I think that it depends on the emission target in terms of spatial and temporal scales (and see what is said on p3 l8-9)

=> the lines p3 l9-12 are quite confusing and may need to be more practical (with practical considerations regarding the temporal and spatial representativity of the aircraft data ?)

This section was poorly worded – in particular we used using a technical term ("over constraining") in a casual way. We've rewritten the end of this paragraph to clarify what we mean. Essentially we are trying to highlight the balance that needs to be struck in allowing the posterior solution to differ from the prior without letting the inversion run out of control (i.e. unrealistically redistributing large fluxes based on a handful of data points).

-> p3 l17: « using a cost function » is a too vague and does not really reflect the difference between the two methods from my point of view (see the comment on this topic in my first review)

-> p3 l20-21: actually the new method does not « adjust » the budget of emissions (as would do a Bayesian approach which gives some statistical weight to the prior information on the targeted emission budget), it derives a new one that is independent of the « prior » budget

=> the lines p3 l17-21 could be improved for both clarity and simplification of the concepts

P3 L17-21 edited to reflect the comments above

- p10 lines 29 to 31: the new « With this caveat » at the beginning, the « are consistent » and then the « For CH4 however » do not really fit well all together, or at least it now confuses me and it requires some thinking.

Section reworked

- p11 l9-15 this is confusing, because it addresses 3,4 general topics together in few lines; some of these topics could rather be discussed at the end of this section or in the last sections of the paper (they are not related to CH4 only)

It is true that these are general points and are not specific to $CH_4$. However, we believe the issues of temporal and spatial representativeness are best discussed in the context of the result for $CH_4$ because we are able to compare and contrast our result with the annual, national verification report (which doesn't cover $CO_2$ or CO). We have edited the lead up to this section to explicitly state that the issues discussed here are applicable to any species, and we have redrafted the specific lines referred to (l9-15) to remove any confusion.

- p11 l23: I think it's more about the diff between the budget of NPP in the footprint of the measurements used to compute the London emissions (both outside and inside London) vs that in the footprint of the background concentrations than about the diff between the NPP in London and outside London

Agreed – statement added to this effect.

-p14l30 to p15 l7: this paragraph should be improved, its lack of structure and clarity makes it difficult to understand

Paragraph redrafted

- some examples of sentences that could be revised and better combined to sentences before and after: p2 l16-17, p2 l33 to p3 l2 (going around in circle), p6 l7 to 9 (going around in circle, and the grid is defined in the next sentence), p11 l9 (unclear), p11 l12 (wind bias) vs l13 (random error in dispersion modelling), p14l21-23 (« in that study … in that study »)

All revised

- the comparison between the modeled and measured wind directions is discussed in the answer to the reviews, but not in the text while I feel that it is an interesting analysis. The bias at the measurement locations may not reflect that on the average wind direction between London and this

transect, but it is important to show that the author did not miss an easy explanation for the mismatch between the main patterns of the modeled and measured plumes or backgrounds.

A discussion of this is included in Sect. 3.1.2 (p10 l4-10 in the redrafted version) – we have expanded this slightly to incorporate elements that were previously only in the author response.

[revised manuscript text omitted]

---

## Author Response (AR3)

**Author Response**

We'd like to thank the editor for catching this issue – we agree that the mole density should be that at the emission location not the sampling location. We have re-run the model to produce the appropriate output to enable this calculation and repeated our analysis. The corrected results have been updated in the text and tables, and we have replotted Figs. 2, 3, 4 and 6 with the new simulated data. The changes do not affect the conclusions so the only significant textual change is in the description of the model run itself.

[revised manuscript text omitted]
$^{-1}$) | | Ratio | Mean flux density ($\mu$mol m$^{-2}$ s$^{-1}$) | | Ratio | Mean flux density ($\mu$mol m$^{-2}$ s$^{-1}$) | | Ratio |
| | Measured | Simulated | | Measured | Simulated | | Measured | Simulated | |
| 287 | 1.79 | 1.74 | 1.03 | 2.70 | 3.41 | 0.79 | 526 | 330 | 1.60 |
| 460 | 1.65 | 1.79 | 0.92 | 2.12 | 3.22 | 0.66 | 468 | 331 | 1.41 |
| 575 | 1.94 | 1.67 | 1.16 | 1.96 | 2.91 | 0.67 | 556 | 300 | 1.85 |
| **Overall** | 1.79 | 1.73 | **1.03** | 2.28 | 3.19 | **0.71** | 516 | 321 | **1.61** |

Table 2: Mean flux densities calculated using the flux-dispersion method, given for each transect and taken over all three transects. The ratios between measured and simulated flux densities are all given.

|  | CO | | CH$_4$ | | CO$_2$ | |
|---|---|---|---|---|---|---|
|  | Mean | 1σ | Mean | 1σ | Mean | 1σ |
| Flux (kmol s$^{-1}$) | 0.178 | 0.006 | 0.182 | 0.009 | 44.7 | 1.2 |
| NAEI emissions (kmol s$^{-1}$) | 0.079 | N/A | 0.149 | N/A | 14.5 | N/A |
| **Ratio** | **2.27** | 0.07 | **1.22** | 0.06 | **3.08** | 0.08 |

**Table 3: Bulk fluxes calculated using a conventional mass balance technique and corresponding NAEI emissions, aggregated over the Greater London administrative region. The ratio of mass balance flux to NAEI emission is also given.**

[Figure]

**Figure 1: Aircraft flight track on 04 March 2016, coloured by altitude. Wind barbs are used to represent wind speed and direction, averaged over 5 minutes, using the convention where each full wind barb represents a wind speed of 10 knots. The border of the Greater London administrative region is shown in grey for reference.**

[Figure]

**Figure 2: Altitude-latitude projections of measured mole fraction (a – c) and simulated (d – f) mole fraction enhancement downwind of London for each species.**

[Figure]

**Figure 3: Altitude-latitude projection showing the influence of London on the downwind sampling, as determined from the NAME air histories. The colour scale represents the fraction of aggregated NAME air history $D_{ij}$ within the Greater London administrative region for each NAME release period. Background periods, where the London fraction is less than 0.05%, are shown in red.**

[Figure]

**Figure 4: NAME air histories aggregated over (a) background sampling periods, and b) in-plume sampling periods, overlaid on an NAEI emissions inventory map for CH₄ (shown using a saturated colour scale). Both air histories have been normalised such that they sum to 1, with grey and pink contours shown in each plot surrounding the vast majority (99.9995%) of sample influence. These contours are included in panel c), d) and e) to provide a better visual comparison between the two aggregate air histories in the context of the inventory emissions for CH₄, CO₂ and CO respectively.**

[Figure]

**Figure 5: Comparison between wind speeds measured by the aircraft and the corresponding wind speeds at the aircraft location from the UKV model. It can be seen that the model generally overestimates wind speed within the boundary layer.**

[Figure]

**Figure 6: Measured and simulated flux densities for $CO_2$, $CH_4$ and CO, given for the three transects (287 m, 460 m and 575 m) used to assess inventory accuracy.**

[Figure]

**Figure 7: Altitude-latitude projections of: a) – c) measured data, d) – f) kriged data, g) – j) kriging standard error, for CO₂, CH₄ and CO respectively.**